# VideoFlexTok: Flexible-Length Coarse-to-Fine Video Tokenization

Andrei Atanov [* 1 2]   Jesse Allardice [* 1]   Roman Bachmann [2]   Oğuzhan Fatih Kar [1 2]
Devon Hjelm [1]   David Griffiths [1]   Peter Fu [1]   Afshin Dehghan [1]   Amir Zamir [2]

https://videoflextok.epfl.ch

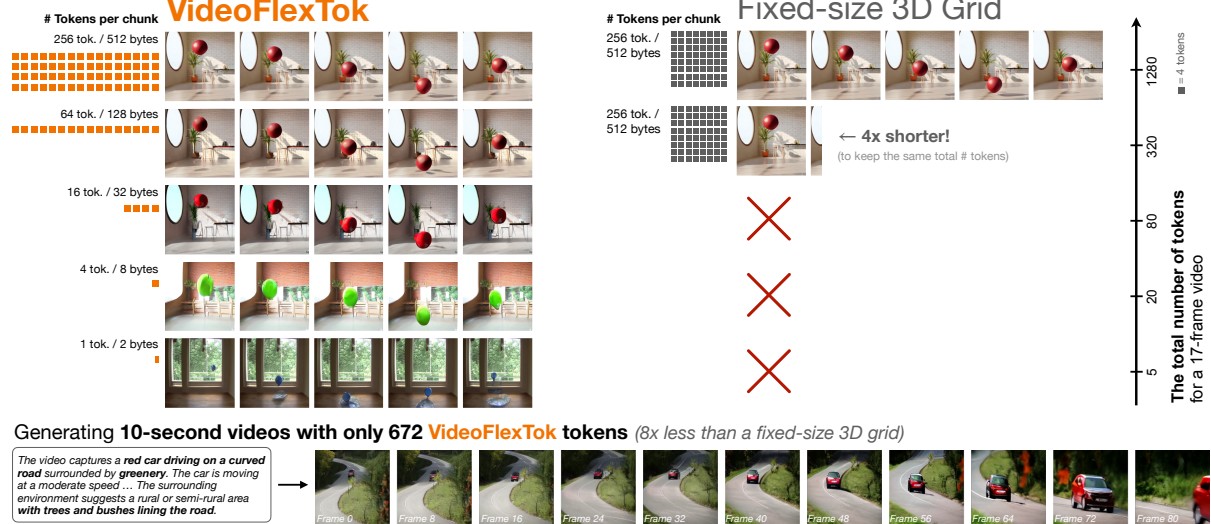

*Figure 1.* **VideoFlexTok represents videos with a flexible-length coarse-to-fine sequence of tokens.** *Top:* Compared to the common 3D grid tokenizers, which can adjust the token sequence length only by reducing the video length, VideoFlexTok can represent the same-length video with a *varying number of tokens corresponding to different levels of detail* – with just a few tokens *emergently* capturing abstract information, such as semantics and motion. *Bottom:* This property enables efficiency, demonstrated here by training a text-to-video model to generate 10-second 81-frame videos using just 672 tokens; 8× fewer than the 5376 required by a comparable 3D grid tokenizer (NVIDIA et al., 2025; Tang et al., 2024; Yu et al., 2024a).

## Abstract

Visual tokenizers map high-dimensional raw pixels into a compressed representation for downstream modeling. Beyond compression, tokenizers dictate what information is preserved and how it is organized. A *de facto* standard approach to video tokenization is to represent a video as a spatiotemporal 3D grid of tokens, each capturing local information from the original signal. This requires the downstream model, e.g., a text-to-video model, to learn to predict all low-level details "pixel-by-pixel" irrespective of the video's inherent complexity, leading to high learning complexity. We present VideoFlexTok, which represents videos with a *variable-length sequence of tokens structured in a coarse-to-fine manner*, where the first tokens (emergently) capture information such as semantics and motion, and later tokens add fine-grained details. The generative flow decoder enables realistic video reconstructions from any token count. This representation structure allows adapting the token count to downstream needs and encoding videos longer than the baselines within the same budget. We evaluate VideoFlexTok on class- and text-to-video generative tasks and show that it yields more efficient training than 3D grid tokens, achieving comparable generation quality (gFVD and ViCLIP Score) with a 5x smaller model (1.1B vs 5.2B). Finally, we show how VideoFlexTok can enable long video generation without prohibitive computational cost by training a text-to-video model on 10-second 81-frame videos with only 672 tokens, 8x fewer than a comparable 3D grid tokenizer.

*Equal contribution [1]Apple [2]Swiss Federal Institute of Technology Lausanne (EPFL). Correspondence to: Andrei Atanov <andrei.atanov@epfl.ch>.

*Proceedings of the 43rd International Conference on Machine Learning*, Seoul, South Korea. PMLR 306, 2026. Copyright 2026 by the author(s).

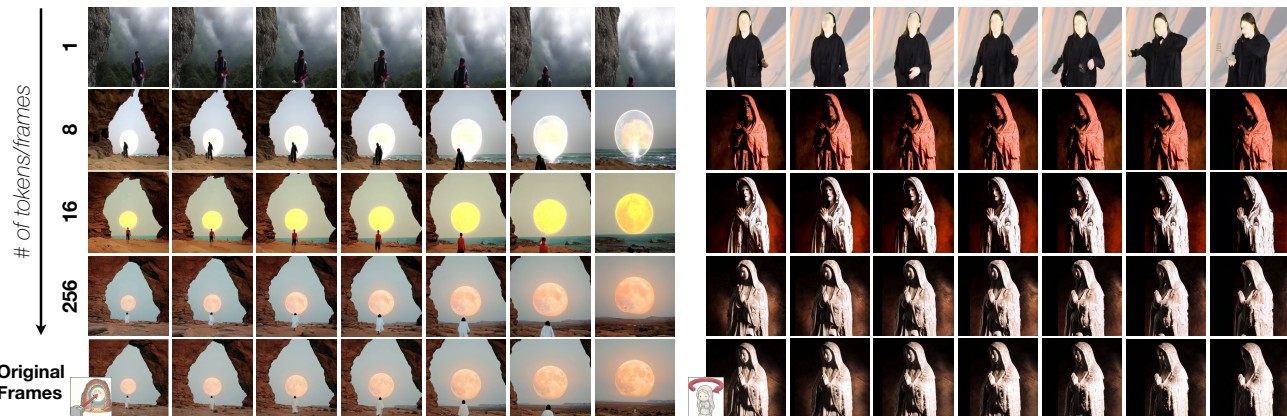

*Figure 2.* **VideoFlexTok reconstructions from a variable number of tokens.** We find that just a few `VideoFlexTok` tokens capture information such as the semantic identities (e.g., a woman in the right example), scene geometry (the *"arch"*), camera motion (moving forward), and object motion (rotation).

## 1. Introduction

Video modeling[1] is computationally expensive, primarily triggered by the high dimensionality of the raw pixel signal that increases with the length of the video (OpenAI, 2025; 2024; DeepMind, 2024; Wan et al., 2025; HaCohen et al., 2024; Kong et al., 2024; Yang et al., 2025; Kondratyuk et al., 2024). Visual tokenization aims to alleviate this problem by compressing the raw visual signal into a lower-dimensional latent space (Rombach et al., 2022; Van Den Oord et al., 2017; Esser et al., 2021).

Beyond just compression, however, tokenizers also define *what* information is preserved and *how* it is structured within the representation. These are important properties that influence the downstream performance (Ramanujan et al., 2025; Zheng et al., 2025). Most common video tokenizers structure their representations as a fixed-size spatiotemporal 3D grid of tokens, each corresponding to some local information in the original signal (NVIDIA et al., 2025; Tang et al., 2024; Yu et al., 2024a). As a result, any video is represented by the same number of tokens regardless of the complexity of its content. In addition, most tokenizers aim for accurate reconstruction, thereby prioritizing the preservation of pixel-level details in their representations. Therefore, a downstream, e.g., text-to-video, generative model that consumes the tokens needs to learn to simultaneously predict low-level details as well as the more abstract structure, such as the overall semantics and motion. This leads to unnecessary computational cost.

This work presents `VideoFlexTok`, a video tokenizer that *represents videos with a flexible-length sequence of tokens ordered in a coarse-to-fine manner*. The first tokens *emergently* encode the most salient semantic, geometric, and motion information, and later tokens add fine-grained details. `VideoFlexTok`'s decoder is a generative rectified flow model that can produce realistic videos given any number of tokens. This representation structure enables *adaptively and drastically*[2] reducing the signal dimensionality while preserving useful abstract information (see Figures 1 and 2).

We evaluate `VideoFlexTok` on class-to-video and text-to-video downstream tasks. We show that, compared to *de facto* standard fixed-size 3D grid tokenizers, using `VideoFlexTok` results in much lower downstream computational cost. For example, we find that one can achieve a comparable level of performance with an order of magnitude less compute (Figure 6). Finally, we demonstrate how these properties can enable modeling videos of longer duration without extensively increasing the computational cost. Specifically, we train a text-video model directly on 10-second videos represented with only 672 tokens, $8\times$ fewer than standard 3D grid tokenizers, yet capturing most useful semantic and motion information (see Figure 1).

## 2. Related Work

**VAE-based grid tokenization.** VAE and VQ-VAE autoencoders have become a *de facto* standard approach to visual tokenization (Van Den Oord et al., 2017; Esser et al., 2021; Kingma & Welling, 2013). They encode the original pixels into a compressed, fixed-size representation, preserving the original signal's structure. This approach is widely applied across different visual domains (Mizrahi et al., 2023; Chang et al., 2022; 2023; Li et al., 2023b; Villegas et al., 2023; Hu et al., 2023), with (Junke et al., 2024; Lu et al., 2025;

---

[1]By "video models" we refer to a broad class of models that have video as an output (potentially, represented by a latent space), including, e.g., text-to-video, image-to-video, and world models.

[2]Up to 256× fewer bytes per frame compared to most standard discrete video tokenizers (Tang et al., 2024; NVIDIA et al., 2025; Yu et al., 2023).

Ma et al., 2025) developing unified tokenizers across multiple modalities. Ge et al. (2022); Yu et al. (2023; 2024a); Yan et al. (2021); Li et al. (2025); Tang et al. (2024) adopt similar techniques, compressing videos both spatially and temporally into a spatiotemporal 3D token grid. In contrast, `VideoFlexTok` resamples the original video signal into a variable-length coarse-to-fine sequence of tokens not tied to local patches. This enables representing the underlying signal at varying levels of detail depending on its inherent complexity and downstream needs.

**1D and semantic tokenization.** More recent works rethink the standard VAE-based grid tokenization by resampling images and videos into compact *1D* sequences (Yu et al., 2024b; Wang et al., 2025), enabling *flexible-length* tokenization (Bachmann et al., 2025; Duggal et al., 2025; Yan et al., 2025; Miwa et al., 2025; Wang et al., 2024a; Wen et al., 2025), or introducing a *semantic bias* during tokenization training (Hu et al., 2023; Ma et al., 2025; Lu et al., 2025; Yu et al., 2025). `VideoFlexTok` adopts resampling and variable-length tokenization from FlexTok (Bachmann et al., 2025), and DINO-based semantic bias (Yu et al., 2025), combining and tailoring these components to the video domain. Unlike ElasticTok (Yan et al., 2025), our tokenizer achieves a much higher compression rate, and we demonstrate its benefits beyond compression, showing, for example, its compute-efficiency in downstream video generation tasks.

**Video modeling in abstract spaces.** Different from modeling in reconstruction-based spaces, Assran et al. (2025); Zhou et al. (2025); Walker et al. (2025) explore learning temporal dynamics in more abstract spaces, e.g., semantic features of pre-trained vision models (Oquab et al., 2024; Radford et al., 2021; Zhai et al., 2023) or down-sampled VAE latents (Jin et al., 2025). More recently, Yin et al. (2025); Li et al. (2025) show that a hierarchical approach of predicting first abstract tokens and then decoding them into the pixel space leads to more efficient image and video modeling. Our work contributes to this area by developing a distinct flexible representation with a coarse-to-fine structure that can vary its compactness level, adapting to specific downstream needs, unlike commonly adopted fixed-sized representations from pre-trained off-the-shelf models.

## 3. Method

We start by describing the properties we want to incorporate into `VideoFlexTok`, motivating the particular design choices we make. In the subsequent sections, we describe our method in more detail (see Figure 3 for an overview).

`VideoFlexTok` is an autoencoder. It *encodes a video into a flexible-length sequence of tokens[3] representing it in a coarse-to-fine manner.* We follow (Bachmann et al., 2025)

---

[3]We use tokens and representations interchangeably.

and use encoder with register tokens (Darcet et al., 2024; Yu et al., 2024b) that resamples the original spatiotemporal video grid into a two-dimensional structure, where the first dimension corresponds to time and the second to the coarse-to-fine structure. To induce the coarse-to-fine hierarchy along the second dimension, we use nested dropout (Kusupati et al., 2022; Wang et al., 2024a; Bachmann et al., 2025), which drops a random number of register tokens from the end. While this alone induces the structure, reconstruction-focused objectives tend to prioritize low-level details (Van Den Oord et al., 2017; Esser et al., 2021; Rombach et al., 2022), preventing first tokens from capturing semantically meaningful information. We, therefore, use a semantic bias by distilling features from a pre-trained vision encoder (Hu et al., 2023; Bachmann et al., 2025; Ma et al., 2025). Note that *no direct supervision is applied as to what information should be encoded in each level of hierarchy, which is purely emergent* through the variable compression mechanism. In addition, since we use DINOv2 (Oquab et al., 2024) as the vision encoder, which is trained in a self-supervised way, no semantic label supervision is applied through it. This first property enables representing videos with a varying levels of detail, which can be adapted to specific downstream needs (see Figures 1 and 2).

Second, the decoder converts any number of tokens into a realistic, plausible video. To enable this, we train the decoder as a generative flow-based model (Bachmann et al., 2025; Ge et al., 2023). In our evaluations, we show that this property allows us to train a downstream conditional generative model, e.g., text-to-video, to produce shorter token sequences that focus on the most relevant information and reusing the decoder to map them into the pixel space, considerably reducing training cost while still producing realistic samples that align with the given conditioning.

### 3.1. Encoding videos into flexible-length representation

Given the 3D spatiotemporal video VAE latents (Tang et al., 2024) $p \in \mathbb{R}^{T \times HW \times D}$ (after flattening along the spatial dimension), and learnable register tokens $r \in \mathbb{R}^{T \times K \times D}$ (Darcet et al., 2024), we construct the input sequence by interleaving them along the temporal dimension $[p_1, r_1, \ldots, p_T, r_T]$. We operate in the VAE latent space mainly to reduce the cost of training the flow decoder (Rombach et al., 2022). We refer to $p_t$ as a latent frame and to $K$ as *the maximum number of tokens per latent frame*. We then pass this sequence through the encoder with the time-causal attention mask, where the tokens $\{p_t, r_t\}$ for each latent frame can only attend to the past latent frames $\{p_i, r_i\}_{i<t}$. In contrast to Wang et al. (2025), which uses full self-attention and represents videos as a flat 1D sequence, our design preserves the time-causal structure of the original signal. This design enables streaming-compatible tokenization, where frames are processed sequentially without

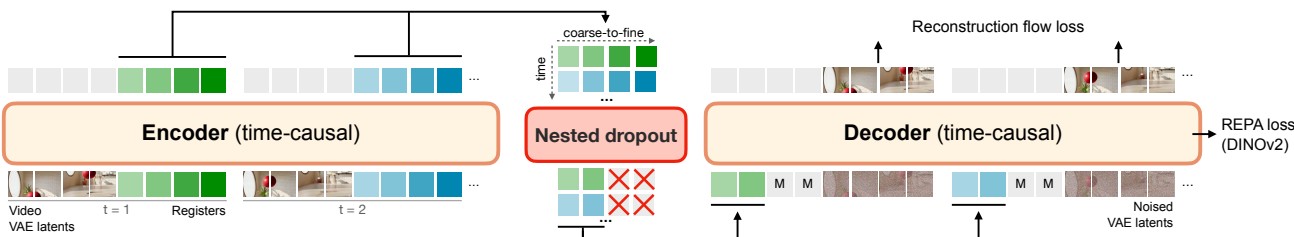

*Figure 3.* `VideoFlexTok` **overview.** *The encoder* takes the spatiotemporal VAE video latents, interleaves them with learnable register tokens across the time dimension, and passes them through the Transformer with a time-causal attention pattern. This results in a *2D representation with the temporal and coarse-to-fine dimensions. Nested dropout* randomly drops a random number of last register tokens along the 2nd dimension, inducing the coarse-to-fine structure. *The decoder* is a flow-based generative model that temporally interleaves masked register tokens with noisy VAE latents and passes them through a time-causal Transformer. *The losses* are: 1) the rectified flow reconstructive loss that predicts clean patches from their noised version and tokens, and 2) the representation alignment loss (Yu et al., 2025) between the decoder and DINOv2 (Oquab et al., 2024) features, which distills semantic information into `VideoFlexTok` tokens.

requiring access to future frames, and improves downstream generation performance, as we find in Section 4.5. In addition, we follow (Bachmann et al., 2025) and use a causal self-attention mask within the register tokens, and we did not find alternative masking patterns to improve performance.

Since our downstream architecture is an autoregressive GPT-like Transformer with cross-entropy loss, we apply FSQ (Mentzer et al., 2024) quantization (64000 codebook size) to the register tokens' output for discretization and use it as the video representation, denoted as $\hat{r}$. Finally, before decoding, we apply nested dropout (Kusupati et al., 2022) to their second dimension. Specifically, we randomly choose $1 \leq k \leq K$ and mask the last $K - k$ tokens for each $\hat{r}_t$.

### 3.2. Generative decoder with semantic bias loss

After the encoder, we interleave the masked registers $\hat{r}$ with the noised VAE latents $\tilde{x} = \alpha \cdot \epsilon + (1 - \alpha) \cdot x$ along the time dimension $[\hat{r}_1, \tilde{x}_1, \ldots, \hat{r}_T, \tilde{x}_T]$, pass them through the DiT decoder (Peebles & Xie, 2023), and apply a rectified-flow loss (Liu et al., 2023). In addition to the flow objective, we add a semantic bias loss in the form of REPA (Yu et al., 2025), which adds a small readout head to an (early) layer of the decoder to predicts self-supervised DINOv2 features and applies a cosine-similarity loss. While originally proposed to improve the diffusion decoder's training efficiency, previous work (Bachmann et al., 2025; Wen et al., 2025), as well as our early experiments (see Section B.1), suggest that it leads to more semantically aware representations in the encoder-decoder architecture. Our final objective, therefore, is as follows: $\mathcal{L}(\theta) = \mathcal{L}_{\text{Flow}} + \lambda \cdot \mathcal{L}_{\text{REPA}}$, where $\theta$ includes the parameters of the encoder, decoder, the REPA head, and the register token queries for the encoder.

In addition to the REPA objective, we use time-causal attention mask in our decoder, which, together with the time-causal encoder and nested dropout, results in a predictive self-supervised objective. Indeed, each $\hat{r}_t$ needs not only to encode information useful for reconstructing the current

frame $p_t$ but also for *predicting* all future frames $\{p_i\}_{i>t}$, which was found to be an effective self-supervised objective (Tong et al., 2022; Bardes et al., 2025; Rajasegaran et al., 2025). Sections B.2 and 4.5 show that this design choice leads to better downstream generative performance compared to the full attention decoder. To benefit from both the improved token structure induced by the time-causal decoder and better reconstruction fidelity of the full-attention decoder (see Table 3), we introduce an additional training stage where we keep the encoder and, hence, the representations fixed and fine-tune the decoder with full attention as described in Section F.

### 3.3. Downstream autoregressive generation

We evaluate `VideoFlexTok` on conditional video generation tasks. Specifically, we follow (Yu et al., 2024a; Wang et al., 2025; Bachmann et al., 2025; NVIDIA et al., 2025) and train a GPT-like conditional autoregressive Transformer for class-to-video (C2V) and text-to-video (T2V) tasks. Importantly, we focus not only on the overall fidelity of the generated samples, commonly measured using FVD (Unterthiner et al., 2018), but also measure how well the model solves the conditioning task (as described in Section 4.1), highlighting the semantic properties of our tokenizer.

### 3.4. Long video tokenization and generation

How can we extend a tokenizer to handle videos longer than it was trained on? One of the main challenges is preserving temporal consistency as we decode subsequent video chunks. This challenge is especially pronounced when decoding from a few `VideoFlexTok` tokens. In this case, the decoder needs to "fill-in" details not present in the tokens, which should be preserved over time as we decode future chunks. Similar to Li et al. (2025), we use the following approach. First, we split a video into fixed-length chunks with $n$ overlapping frames and encode each independently. During decoding, we decode the first chunk as

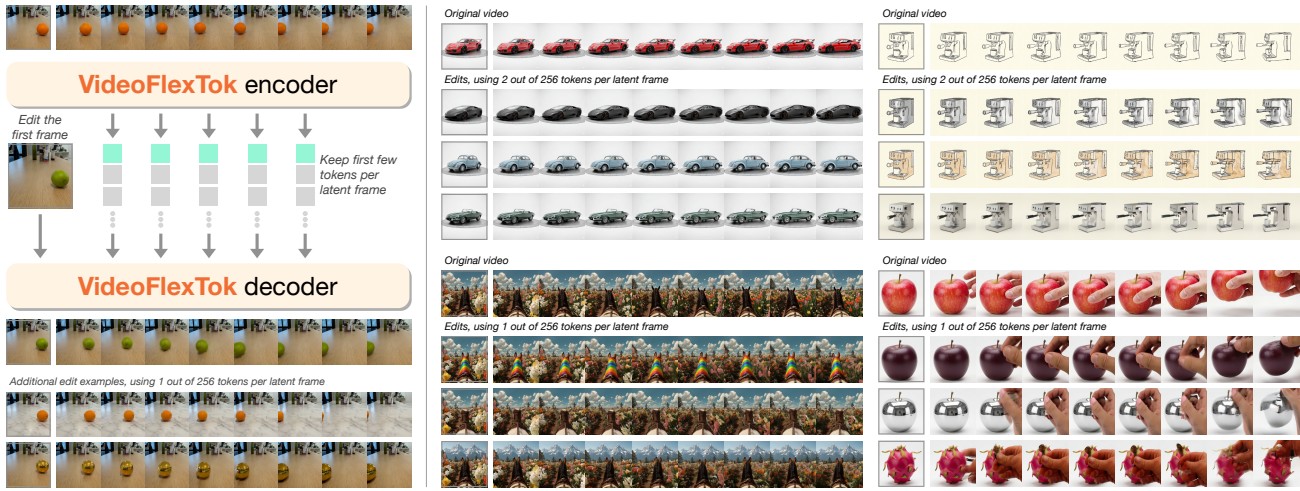

*Figure 4.* **Probing the first `VideoFlexTok` tokens.** We design the following probing experiment to analyze the information contained in the first `VideoFlexTok` tokens. Given a source video, we keep only one or two tokens per latent frame and make an isolated change to its first frame, e.g., changing an orange to an apple, using Nano Banana (Google, 2025). We then condition the decoder on both the original tokens and the new edited frame for reconstruction. We find that, in most cases, `VideoFlexTok` preserves the motion pattern from the original video and visual appearance from the edited frame throughout the reconstructed video, *suggesting that the first tokens primarily capture the motion information.*

usual, and for subsequent chunks, we condition the flow decoder on the last $n$ generated frames. This allows us to preserve temporal consistency in the decoded video even when reconstructing from a few tokens (see Figures 1, 4 and 8).

During downstream generative modeling, this `VideoFlexTok`'s design and aforementioned properties enable the downstream AR transformer to model longer-range temporal dependencies without extensively increasing its context length. Specifically, we can now train the AR model to predict only the first few tokens per latent frame capturing the most essential information and use `VideoFlexTok` to decode it back into a consistent video. In Section 4.4, we explore this design and provide a qualitative example. We use 32 tokens per frame, allowing the AR model to generate a 10-second video using only 672 tokens while still expressing the conditioning well (see Figures 1 and 8). For calibration, an off-the-shelf tokenizer (Yu et al., 2024a; NVIDIA et al., 2025; Junke et al., 2024) would require 5376 tokens, extensively increasing both the training and inference cost. This essentially enables to keep longer videos in the context of the AR transformer using a lower token budget.

# 4. Experiments

## 4.1. Implementation details

**`VideoFlexTok` architecture.** We train our tokenizer in the VAE latent space to reduce the computational cost of training the generative flow decoder (Rombach et al., 2022).

We use VidTok 3D VAE (Tang et al., 2024), that maps the original video of shape $(T + 1) \times H \times W \times 3$ to $(\frac{T}{f_t} + 1) \times \frac{H}{f_h} \times \frac{W}{f_w} \times C$. We use the version with $C = 16$ channels and $f = (4, 8, 8)$, which we found to provide a good balance between compactness and reconstruction fidelity. We use a total of 256 registers per each (latent) frame. We parametrize the encoder and decoder Transformer shapes as $\mathrm{depth} = d$, $\mathrm{width} = 64d$, $\mathrm{num\_heads} = d$ following (Tian et al., 2024). For Kinetics-600, we train the tokenizer with $d_{\mathrm{enc}} = d_{\mathrm{dec}} = 18$. For Panda, we use $d_{\mathrm{enc}} = 18$, $d_{\mathrm{dec}} = 28$ and apply additional $[1, 2, 2]$ patchification of the VAE latents to reduce the sequence length.

**AR model training.** We employ a LLaMa-like Transformer (Touvron et al., 2023; Sun et al., 2024) as our autoregressive generative model. For class-conditional generation, we add a class embedding to the `[SOS]` token. For text-to-video generation, we use T5 (Raffel et al., 2020) as the text encoder and add cross-attention layers to the autoregressive Transformer following (Sun et al., 2024; Kondratyuk et al., 2024). We use the time-first order for `VideoFlexTok`, i.e., we predict the first token for each timestep, then the second and so on, which provides the best overall performance (see Section B.3). We use standard raster-scan order for the 3D grid baseline.

To study AR scaling on `VideoFlexTok` tokens, we follow a compute-aware procedure inspired by Chinchilla-style optimal training (Hoffmann et al., 2022). For our data-rich text-to-video settings, we scale the model size $N$ and training tokens $D$ jointly using the heuristic $D \approx 20N$, and increase the batch size sublinearly with $D$ following a

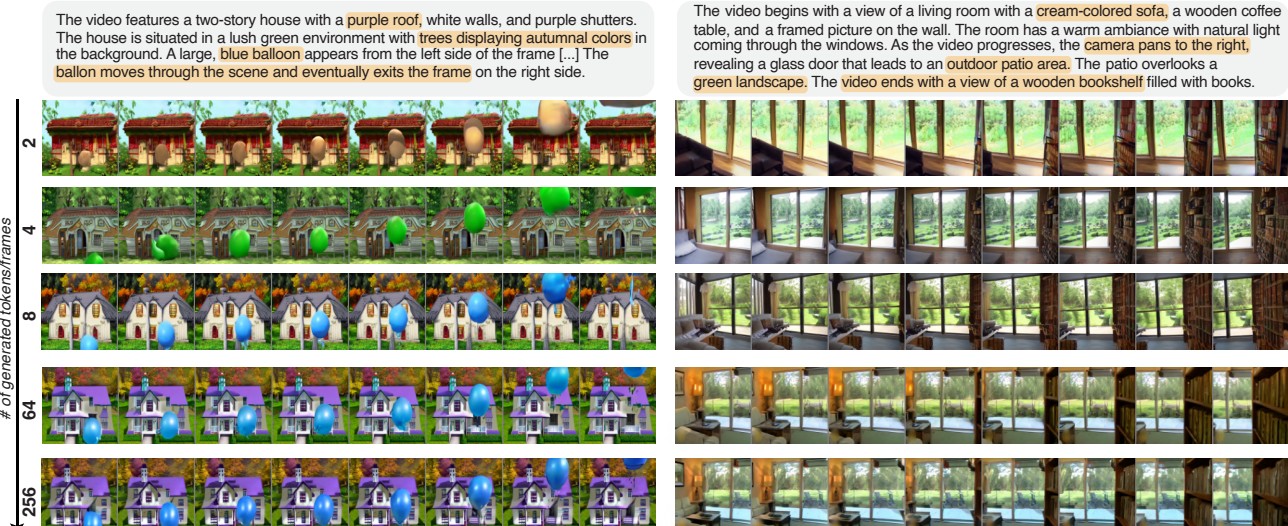

*Figure 5.* **Flexible-length autoregressive text-to-video generation.** A text-to-video generative model using `VideoFlexTok` tokens can generate token sequences of varying length for a given conditioning. All token budgets lead to plausible generations, with 2-4 tokens/frame capturing the overall scene details and motion described in the text conditioning well (e.g., the balloon movement), while generating more tokens can express more fine-grained details (e.g., the number of floors).

square-root power law to remain within the optimal training regime (Zhang et al., 2025). This defines a FLOPs sweep from $1.6 \times 10^{20}$ to $5 \times 10^{21}$ for models from 400M to 5.2B parameters. In the data-limited class-to-video setting, we, instead, fix either $D$ or $N$ and vary the other parameter.

**Datasets.** For class-to-video generation, we use videos from the Kinetics-600 (Kay et al., 2017; Carreira et al., 2018) dataset at a resolution of $128 \times 128$. For text-to-video generation, we use a subset of Panda70M (Chen et al., 2024b) with detailed synthetic captions generated following (Chen et al., 2024a), and use a resolution of $256 \times 256$. For both datasets, we extract 17-frame 4-second clips during training of both the tokenizer and autoregressive models, except in Section 4.4, where we use 81-frame 10-second videos for the autoregressive model training. Note that in both cases, we model substantially longer videos than the more standard ∼0.5 seconds (Wang et al., 2025; Yu et al., 2023).

**Evaluation metrics.** We focus our evaluations on two aspects, fidelity and conditioning alignment. We use Frechét Video Distance (FVD) (Unterthiner et al., 2018) for both generation (gFVD) and reconstruction (rFVD) fidelity. We follow VBench (Huang et al., 2024) and use a UMT-L (Li et al., 2023a) model finetuned for Kinetics-600 classification to measure class-video alignment (using the first 16 out of 17 frames), and ViCLIP-InternVid-10M-FLT (Wang et al., 2024b) to measure text-video alignment (subsampling 8 out of 17 frames using a temporal stride of 2). In both cases, we interpolate the videos to 224x224 resolution.

We provide more implementation details in Section F.

### 4.2. Flexible-length tokenization and generation

**Tokenization.** Figures 1 and 2 show that `VideoFlexTok` can represent videos in a coarse-to-fine manner with the flow decoder producing plausible generations based on any number of tokens. Importantly, we find that first tokens in the hierarchy capture semantically-meaningful information, such as object type, their motion and overall scene geometry, while abstracting away more nuanced details, such as color information. In Figure 4, we further probe what type of information is encoded in the first tokens, by conditioning the decoder on 1 or 2 tokens per latent frame from a source video and an edited first frame of the same video. We find that `VideoFlexTok` decoder preserves the visual edits made to the first frame and applies the motion from the source video, e.g., by transforming a rolling orange into a rolling apple. This suggests that the first tokens primarily capture the motion information.

**Generation.** Training an AR model on `VideoFlexTok` tokens naturally leads to a coarse-to-fine autoregressive generation order. Figures 5 and 7 demonstrate how the text-to-video generative model expresses the text conditioning with better precision as generates more tokens. Interestingly, Figure 7 suggests a trade-off in terms of the fidelity between the amount of information generated by the AR model and the flow decoder, suggesting that balancing compute between the AR and flow generative models might be a more efficient strategy.

These results suggest that we can train smaller generative models with fewer training steps by representing videos with shorter sequences, thereby reducing downstream com-

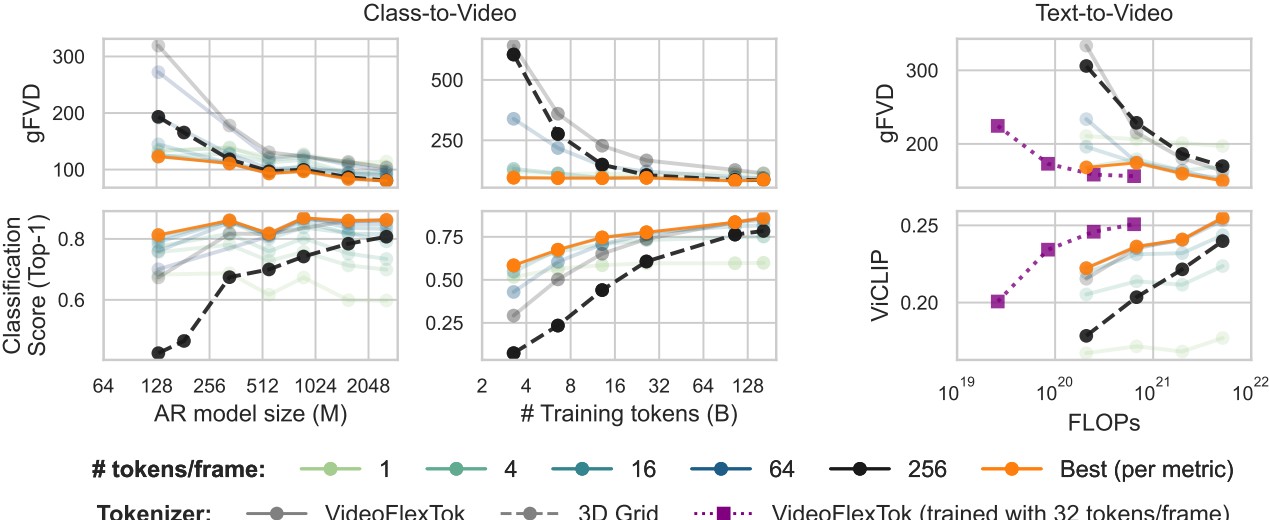

*Figure 6.* **Compute-efficient AR training with `VideoFlexTok`.** We show how the fidelity (top) and alignment (bottom) metrics change across three complementary scaling axes. **Scaling the model size (left).** We show how the fidelity (top, gFVD) and alignment (bottom, Classification Score) metrics change as we scale the size of the class-to-video autoregressive model. Using `VideoFlexTok` maintains good fidelity across a wider range of model sizes while solving the conditioning task well, achieving a much higher alignment score with smaller models. *This implies that we can train much smaller models to solve the class-to-video downstream task.* **Scaling the number of training tokens (middle).** We show how the fidelity (top, gFVD) and alignment (bottom, Classification Score) metrics evolve during training of the class-to-video downstream model. We use the 1.3B AR model size for this experiment. Using `VideoFlexTok` enables having good reconstructions throughout the whole training, and *achieves similar or better alignment using 5–10 times fewer training tokens than the 3D Grid tokenizer.* **Text-to-video FLOPs efficiency (right).** We show how the fidelity (top, gFVD) and alignment (bottom, ViCLIP Score) metrics change as we scale both the model size and the number of training tokens in the compute-optimal way. We vary the model size from 400M to 5.2B with the estimated ratio of $D/N = 20$ (Hoffmann et al., 2022). In addition to models trained on full-length `VideoFlexTok` token sequences, we train a model on shorter 32-token sequences per latent frame while keeping the number of steps the same, resulting in much lower computational cost (purple line). *Overall, we find that autoregressive generative modeling over `VideoFlexTok` tokens can achieve similar performance using an order of magnitude less compute.*

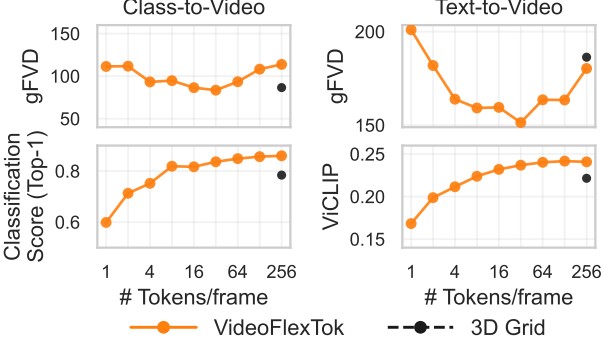

*Figure 7.* **Flexible-length generation.** We measure fidelity (top, gFVD), and alignment (bottom, classification score and ViCLIP similarity, see Section 4.1) for `VideoFlexTok` and 3D grid tokenizers on class-to-video (left) and text-to-video (right) tasks. We use 1.3B and 2.0B AR models for class-to-video and text-to-video, respectively. Using much fewer tokens, `VideoFlexTok` maintains fidelity comparable to or better than the 3D tokenizer, while achieving higher alignment, i.e., better solving the corresponding conditional generation task.

putation costs while achieving similar performance. We demonstrate this quantitatively in the next section.

### 4.3. Downstream efficiency via flexible compactness

**Class-to-video: model size and training time efficiency.** Figure 6 shows that adjusting the number of generated `VideoFlexTok` tokens to the specific downstream needs leads to substantial efficiency gains. Specifically, we find that we can train a 5-10× smaller AR model or use 5-10× fewer training tokens (not to be confused with the sequence length per video) to achieve comparable or better performance than the 3D grid tokenizer, which always needs all tokens to be generated. In addition, we find from the middle plot, showing the metrics' progress as we increase the number of training tokens, that it is not necessary to train different AR models for each specific sequence length. Indeed, in this experiment, we use full sequences during training and find that the model can generate shorter sequences well already early in training. Naturally, alignment performance of short sequences (1-4 tokens/frame) eventually saturates, while the performance of longer sequences (64-256) steadily increases. This allows a practitioner to easily achieve per-

The video captures a person in front of the stove **whisking eggs in a white bowl**. As the video progresses, the person slowly **pours the eggs into the pan** and starts to stir them. **The final image: a cooked omelette in the pan.**

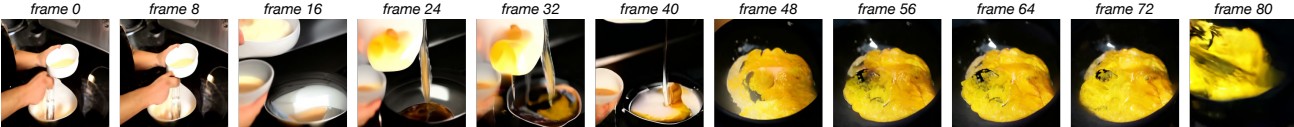

*Figure 8.* **Long text-to-video generation.** We show an exemplar generation of a 10-second 81-frame video using only 672 tokens (32 tokens per frame).

formance better than the fixed 3D grid tokenizer across any compute regime, without retraining the tokenizer or the AR model.

**Text-to-video: FLOPs efficiency.** Since our text-to-video dataset is orders of magnitude larger, we scale both the model size and the number of training tokens in a compute-optimal-inspired way as described in Section 4.1. Similar to the class-to-video results, we find that using `VideoFlexTok` and adjusting the number of the AR-generated tokens achieves performance comparable to the 3D grid counterpart with *an order of magnitude less compute and outperforms it in our largest tested compute regime.* In addition, we train a series of models using shorter sequences (32 tokens/frame) during training, which further significantly reduces the training cost while still achieving comparable performance.

While we focus on analyzing training compute scaling, inference cost is another axis of interest. Indeed, generating fewer tokens with the AR model might require more denoising steps with the flow decoder. In Section E, however, we show that this compute allocation leads to better performance across different inference budgets. In addition, methods that reduce the number of denoising steps can further reduce the inference cost of the flow decoder (Salimans & Ho, 2022; Yin et al., 2024; Lu et al., 2022).

### 4.4. Long video generation

Finally, we provide an example of how the above efficiency gains can enable longer temporal modeling without substantially increasing the computational cost of training. As also described in Section 3.4, we train a text-to-video model on 10-second 81-frame videos ($\sim$ 5x longer than in the previous experiments). Building on our previous findings, we use 32 tokens per frame, resulting in only 672 tokens per video (for calibration, a comparable 3D grid tokenizer would require 5376 tokens). We train a 3.2B model for $\sim$55B tokens, resulting in $\sim 10^{21}$ total FLOPs (the middle range of our scaling experiments). Figures 1 and 8 show exemplar generations. The model can generate coherent, 10-second videos that generally follow the text conditioning, all without exceeding the computational budget and context length of shorter-length models from Section 4.3.

*Table 1.* **System-level comparison on Kinetics-600 class-to-video generation.** For each tokenizer, we evaluate its reconstruction performance using rFVD and its generation quality in terms of fidelity (gFVD) and alignment (Cls. Score) when training a 2.2B class-conditional AR model. For reconstruction, we use 17-frame clips sampled at 4 FPS from the original Kinetics-600 evaluation videos. [†] indicates `VideoFlexTok` results for a sequence of 160 tokens.

| Tokenizer | # Tokens | rFVD ($\downarrow$) | gFVD ($\downarrow$) | Cls. Score ($\uparrow$) |
|---|---|---|---|---|
| VidTok FSQ (Tang et al., 2024) | 1280 | 84.1 | 131.7 | 0.799 |
| Cosmos-DV (NVIDIA et al., 2025) | 1280 | 220.5 | 187.6 | 0.825 |
| Omnitokenizer (Junke et al., 2024) | 1280 | 63.6 | 102.6 | 0.858 |
| LARP (Wang et al., 2025) | 1024 | 42.1 | 87.5 | 0.739 |
| `VideoFlexTok` | 5-1280 | 48.7[†] | 80.0[†] | 0.833[†] |

### 4.5. Additional Results

This section presents ablations on some design choices. Section B provides more results, including the effects of REPA and decoder attention, and comparisons between AR orders.

**Flat 1D vs time-causal 2D registers structure.** Table 2 compares our 2D register design choice, which preserves the time-causal structure of the original video signal, with the 1D flat token structure of LARP (Wang et al., 2025).

*Table 2.* **1D vs 2D registers structure.**

| Register Structure | rFVD ($\downarrow$) | gFVD ($\downarrow$) |
|---|---|---|
| Flat (1D) | **48.9** | 352.1 |
| Time-causal (2D) | 69.9 | **287.6** |

We find that while 1D tokens lead to better reconstruction quality (rFVD) due to the encoder's full self-attention, their downstream generative performance is worse (gFVD), suggesting these tokens are harder to predict, likely due to a lack of sufficient structure.

**Decoder self-attention.** Table 3 compares the full and time-causal decoder self-attention patterns. We find that while full self-attention leads to better reconstruction performance, the time-causal pattern yields better

*Table 3.* **Decoder self-attention pattern.**

| Decoder Attention | rFVD ($\downarrow$) | gFVD ($\downarrow$) 32 Tok |
|---|---|---|
| Full | **58.3** | 211.5 |
| Time-Causal | 80.9 | **175.1** |

downstream generative performance, suggesting that it induces additional useful structure into the tokens. Section B.2 further shows that it also leads to better alignment with only a few first tokens, suggesting that these tokens better capture

semantic information.

**Comparison to off-the-shelf tokenizers.** Table 1 compares `VideoFlexTok` to relevant existing video tokenizers (Tang et al., 2024; Wang et al., 2025; NVIDIA et al., 2025; Junke et al., 2024). For each tokenizer, we train a 2.2B class-to-video AR model for 164B tokens (except LARP, which sees the same number of videos but slightly fewer tokens due to a higher compression rate) and evaluate their reconstruction and generative performance. While using only 160 tokens (6–8× fewer than other tokenizers) during inference, `VideoFlexTok` achieves reconstruction quality (rFVD) comparable to LARP and better generation performance in terms of fidelity (gFVD) and alignment (Cls. Score), except Omnitokenizer, which achieves higher alignment. These results indicate that `VideoFlexTok` is highly competitive under a much tighter token budget

## 5. Conclusion and Discussion

We introduce `VideoFlexTok`, a tokenizer that represents videos with a flexible-length sequence of tokens structured in a coarse-to-fine manner, allowing to adapt these representations to particular downstream needs. Its generative flow decoder can decode realistic videos from any number of tokens. We demonstrate that this structure leads to more computationally efficient generative modeling and can enable the generation of longer videos without substantially increasing the context length and computational cost, effectively democratizing video generative modeling.

We believe that modeling in more compact and semantically-aware abstract representation spaces like `VideoFlexTok` will enable capturing long-range dependencies from videos more efficiently compared to learning them directly from pixels. The coarse-to-fine structure enables capturing the dependencies at different levels of abstractions. This, in turn, can lead to more efficient and performant visual reasoning models that adaptively decide what level of abstraction to work in.

## Acknowledgment

We thank Mingfei Gao, David Mizrahi, Enrico Fini, Philipp Dufter, and Erik Daxberger for their feedback and discussion during the early stages of the project. We also thank Jason Toskov, Rishubh Singh, Kunal Singh, and Ali Garjani for their help in preparing the manuscript. This work was supported under project ID **a08** as part of the Swiss AI Initiative, through a grant from the ETH Domain and computational resources provided by the Swiss National Supercomputing Centre (CSCS) under the Alps infrastructure. This work has received funding from the Swiss State Secretariat for Education, Research and Innovation (SERI).

## Impact Statement

Video research is computationally expensive, preventing compute-constrained parties from entering and contributing to the field. The efficiency implication of `VideoFlexTok` is a major aspect studied in our work. Specifically, `VideoFlexTok` enables representing videos with anywhere between 2 and 256 times less memory than a comparable 3D grid approach, resulting in corresponding downstream efficiency gains. In practice, on common generative tasks, we find that using `VideoFlexTok` can achieve performance similar to that of the 3D grid tokenizer, with 5-10× less computational resources. By developing a more efficient technique for video representation and modeling, our work contributes to the democratization of video research. This has the potential to lower the entry barrier, bringing more parties into the field and diversifying the research landscape.

We use generative modeling as the main testbed for testing `VideoFlexTok`. Our models are not particularly poised for negative use. However, it should be noted that powerful generative models are general tools and can be used in ways the authors did not intend. In addition, the data they are trained on may incorporate societal biases or contain samples collected from the internet in different ways.

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

# A. Additional Qualitative Results

In Figures 15 to 18, as well as in the supplementary archive, we provide additional examples of the variable-length video reconstructions by `VideoFlexTok`.

# B. Additional Ablations

In the ablation experiments, we use the `VideoFlexTok d12-d12` version as described in Table 5 and an autoregressive model with depth 16 and 201M parameters, unless stated otherwise.

## B.1. REPA: semantic inductive bias

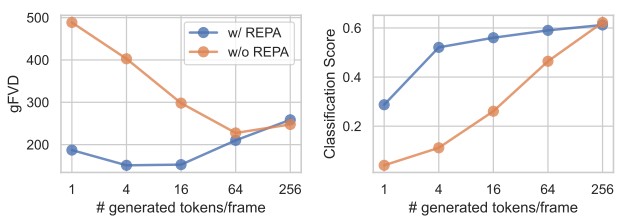

*Figure 9.* **REPA (Yu et al., 2025) loss ablation.** We compare tokenizers trained with and without the REPA loss on the class-to-video downstream task. We find that REPA inductive bias loss improves both the fidelity of the generated samples and the alignment with the class conditioning.

We ablate the use of the additional REPA loss (Yu et al., 2025). Figure 9 shows that using this loss significantly improves the fidelity and alignment score in the few tokens regime.

## B.2. Causal decoder: future prediction pre-training task

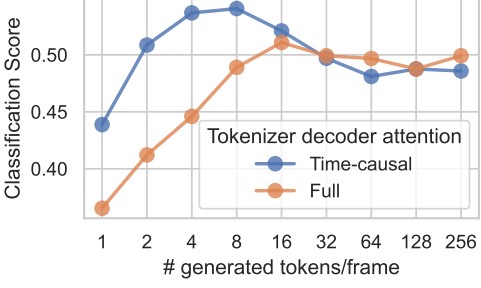

*Figure 10.* **VideoFlexTok decoder attention ablation** We ablate the decoder attention pattern by comparing the alignment score (Classification Score) on the class-to-video generative task. We find that causal attention leads to a higher alignment score with fewer tokens, suggesting the early tokens capture more semantic information in this case (see Section B.2 for discussion).

As described in Section F, we use time-causal attention in the decoder during encoder training. In Section 4.5 and Ta-

ble 3, we demonstrate that this design improves downstream generative performance; thus, we adopt it. As discussed in Section 3.2, this causal design, combined with nested dropout, results in a future prediction task, which was found to be a useful self-supervised pre-training objective (Tong et al., 2022; Bardes et al., 2025; Rajasegaran et al., 2025). We hypothesize, therefore, that this design leads to the first register tokens capturing "more" semantic information. We evaluate this hypothesis by comparing the alignment score for class-to-video downstream generation in Figure 10. We find that using a time-causal decoder yields a higher alignment score with the class label for fewer tokens, suggesting that this information is better captured in this case than with the full attention decoder. The lower classification score with more generated tokens can be explained by the fact that we use a relatively small 200M autoregressive model, which results in poor generation quality for the full sequence length (see, e.g., the trend in Figure 7 where we use a 1.3B AR model and causal decoder).

## B.3. Autoregressive generation order

*Table 4.* **Ablating different autoregressive orders over `VideoFlexTok` tokens.** We compare autoregressive orders on the class-to-video downstream task. We use depth-first and time-first orders for the same underlying tokenizer. We find no significant difference when using all tokens. The time-first order allows adjusting the number of tokens during evaluation, leading to better performance.

| AR order | #Tokens | gFVD ($\downarrow$) |
|---|---|---|
| | 1 | 187.1 |
| | 4 | **151.1** |
| time-first | 16 | 152.7 |
| | 64 | 210.1 |
| | 256 | 246.6 |
| depth-first | 256 | 242.6 |

In this section, we compare the time-first and the depth-first AR orders. Time-first is the default order we use in our experiments in Section 4.3: we first predict the first token across all latent frames, then the second, and so on. In the depth-first order, we predict all tokens for the first latent frame, then for the second, and so on. Table 4 shows the results. We did not find a significant difference between the two AR orders, and the time-first order allows varying the number of generating tokens, which leads to better performance. While it is possible to train an AR model with depth-first order with fewer tokens per latent frame, it requires training a separate model for each token budget. Another approach could be to design a more flexible generative model that can be controlled to produce a different number of tokens. We leave a more extensive exploration of the best way to predict this 2D, coarse-to-fine $\times$ time

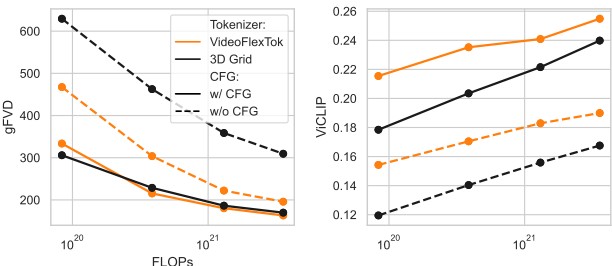

*Figure 11.* **Hierarchical vs. raster-order generation.** We compare the `VideoFlexTok` and 3D grid tokenizers' performance at the same sequence length (1280 tokens). We find that hierarchical generation with `VideoFlexTok` leads to 1) better alignment (ViCLIP score) and 2) much better fidelity (gFVD) when not using classifier-free guidance.

`VideoFlexTok` tokens for future work.

## C. Hierarchical Generation with `VideoFlexTok`

In the main paper, in Section 4.3, we mainly focus on the downstream benefits brought by the flexible compression rate. In this section, we focus on studying the effect of the hierarchical coarse-to-fine autoregressive generation order enabled by `VideoFlexTok`. To this end, Figure 11 compares the performance of `VideoFlexTok` and its 3D-grid controlled counterpart at the same compression rate, i.e., using the same number of tokens per frame $N = 256$ for both. First, we find that `VideoFlexTok` achieves a better text alignment score across all scales. Second, we find that `VideoFlexTok` is much less reliant on classifier-free guidance for generation fidelity, achieving a much lower gFVD without it. Note that both tokenizers benefit from the REPA inductive bias (see Sections B.1 and 3.1), with the only differences being the token structure and the use of nested dropout. We hypothesize that, as with text or class conditioning, which are crucial for achieving high fidelity (see, e.g., (Li et al., 2024)) compared to unconditional generation, hierarchical coarse-to-fine generation with `VideoFlexTok` helps split the problem into a sequence of simpler problems, leading to better overall performance.

## D. Rate-Distortion Trade-off

We show the rate-distortion trade-off in Figure 12. Expectedly, we observe a consistent decrease in reconstruction error as we use more tokens. We make two observations suggesting further directions for improvement. First, for MSE and LPIPS, the trend saturates as we use 128-256 tokens, suggesting we might need longer training or other improvements. Second, it should be possible to achieve flat rFVD across token numbers with a sufficiently good generative decoder as it is a distributional metric, suggesting there

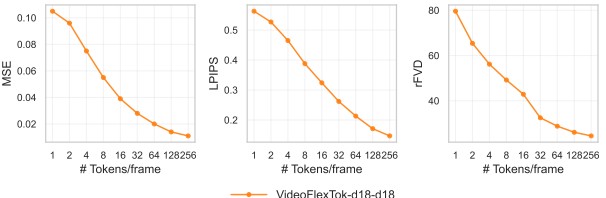

*Figure 12.* **Rate-distortion trade-off.** We show the reconstruction performance, measured using MSE, LPIPS, and rFVD, as a function of the number of tokens used for reconstruction. We use `VideoFlexTok`-d18-d18 and evaluate on Kinetics-600. We observe that using more tokens consistently improves the reconstruction performance.

is room to further improve the decoder.

## E. Inference Cost Analysis

In Section 4.3, we show that using `VideoFlexTok` can drastically reduce the training compute cost and achieve similar performance with smaller models and/or less training. This is mainly achieved by generating fewer tokens with the AR model (though we still see improved alignment even when generating all tokens). This, however, inquires the cost of running the flow decoder for multiple steps to generate the missing details and obtain the final full RGB output (VAE latents in our case). In this section, we study how the performance changes as we scale the inference compute by either sampling more tokens with the AR model or doing more denoising steps with the `VideoFlexTok` decoder for different AR model sizes. We use the `VideoFlexTok d18-d28` tokenizer for this experiment (see Table 5).

Figure 13 shows that for all considered model sizes and inference costs, generating less than 256 tokens per frame (the full sequence) and using the flow decoder achieves a better performance for all inference budgets. In addition, Figure 14 shows inference time estimates for a single H100 GPU, showing an even stronger benefit of using the flow decoder. Note that the wall-clock time depends heavily on the specific setting (hardware, kernels, KV-cache management, batching). Both AR and flow inference can be further optimized, e.g., by flow distillation or MQA/MLA for the AR model. For example, as we increase the batch size, the memory-bound AR model stays at 37 ms/token for batch sizes between 1 and 4, while the compute-bound flow model's cost grows linearly from 123 to 374 ms/step. We use the batch size of 4 in our measurements as it saturates VRAM, but using lower batch size would further skew the trend towards the flow model.

We believe that this trend can be further improved by balancing compute between the AR and flow decoder models more carefully. In addition, the flow inference cost can be significantly reduced by using distillation-based approaches (Sal-

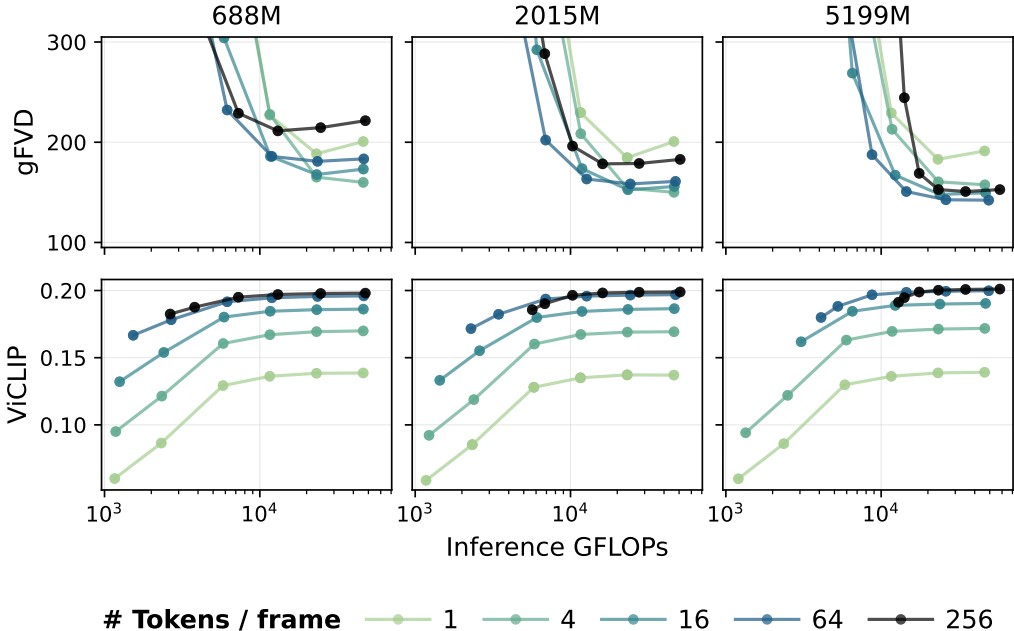

*Figure 13.* **Text-to-video inference cost analysis.** We compare the inference cost of various configurations of the number of AR-generated tokens and the number of `VideoFlexTok` flow decoder steps. For each AR model size and the number of generated tokens, we perform $\{1, 2, 5, 10, 20, 40\}$ denoising steps and plot the corresponding lines. We find that for all considered AR sizes and inference budgets, the best performance is achieved by generating less than 256 tokens per frame and performing multiple denoising steps.

imans & Ho, 2022; Yin et al., 2024; Lu et al., 2022; Zhu et al., 2024). We leave this direction to future work.

## F. Architecture and Training Details

### F.1. `VideoFlexTok` details

We provide a detailed overview of the architecture and training configuration in Table 5. We follow Bachmann et al. (2025) for our overall design and introduce the following changes, extending it to video sequences.

- We extend 1D registers to 2D by adding the time dimension as described in Section 3.1.

- In addition to causal attention over register tokens within a latent frame, we also introduce a time-causal attention mask in both the encoder and decoder.

- We use a pre-trained VidTok video VAE (Tang et al., 2024) with both temporal and spatial compression.

- As the REPA (Yu et al., 2025) head, we use a Transformer with time-causal attention mimicking the decoder design, which we found to perform better in terms of both reconstruction and downstream generation performance in our early explorations.

- We introduce an additional decoder fine-tuning stage where we keep the encoder frozen and make the fol-

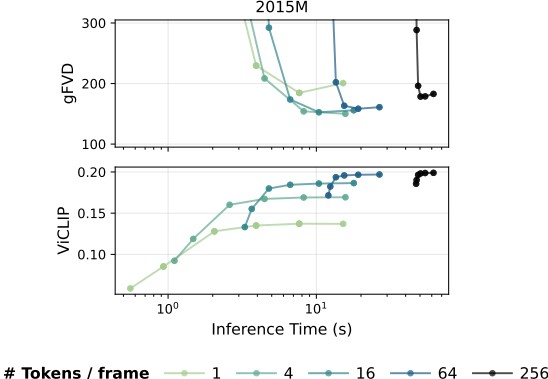

*Figure 14.* **Text-to-video inference time on a single H100 GPU.** In addition to Figure 13, we compare the inference time of various configurations of the number of AR-generated tokens and the number of `VideoFlexTok` flow decoder steps for the 2B AR model. We use a single H100 GPU and use KV-cache for the AR model. For each number of generated tokens, we perform $\{1, 2, 5, 10, 20, 40\}$ denoising steps and plot the corresponding lines. We find that for all considered AR sizes and inference budgets, the best performance is achieved by generating less than 256 tokens per frame and performing multiple denoising steps.

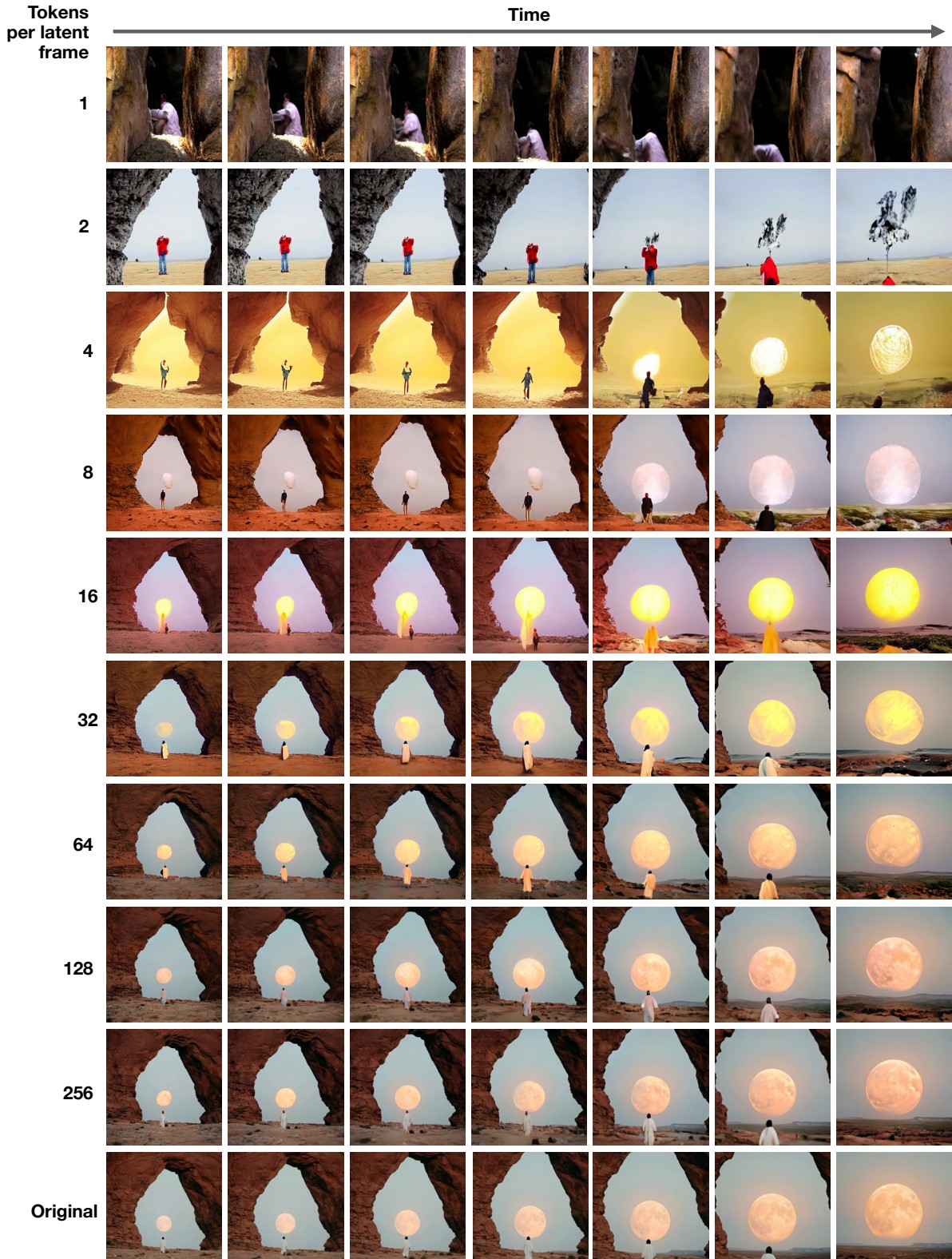

*Figure 15.* **VideoFlexTok reconstruction example**. From top to bottom, each row corresponds to a video reconstructed using 1, 2, 4, ..., 256 tokens. The last row shows the original video.

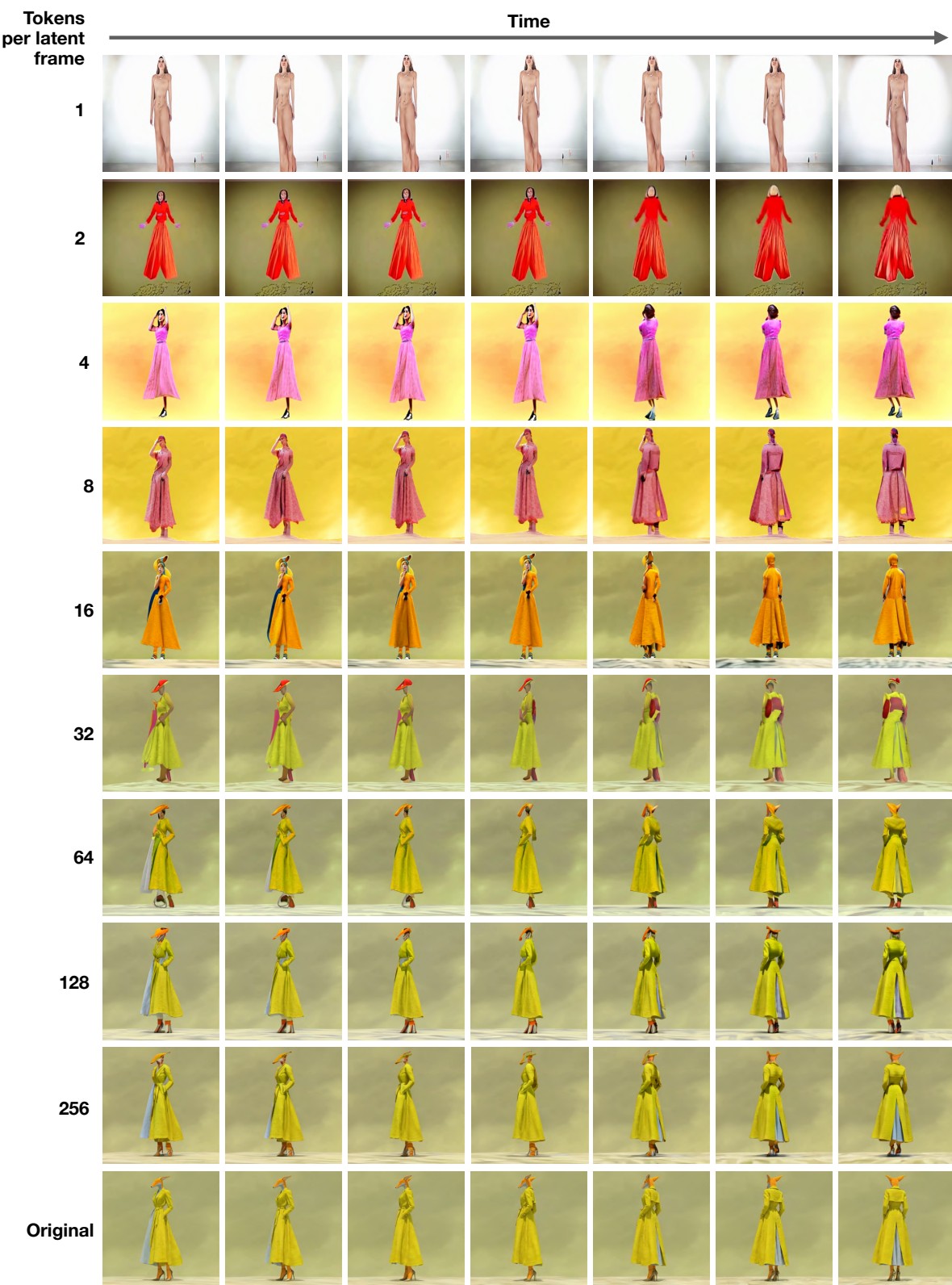

*Figure 16.* `VideoFlexTok` **reconstruction example**. From top to bottom each row corresponds to a video reconstructed using 1, 2, 4, . . . , 256 tokens. The last row shows the original video.

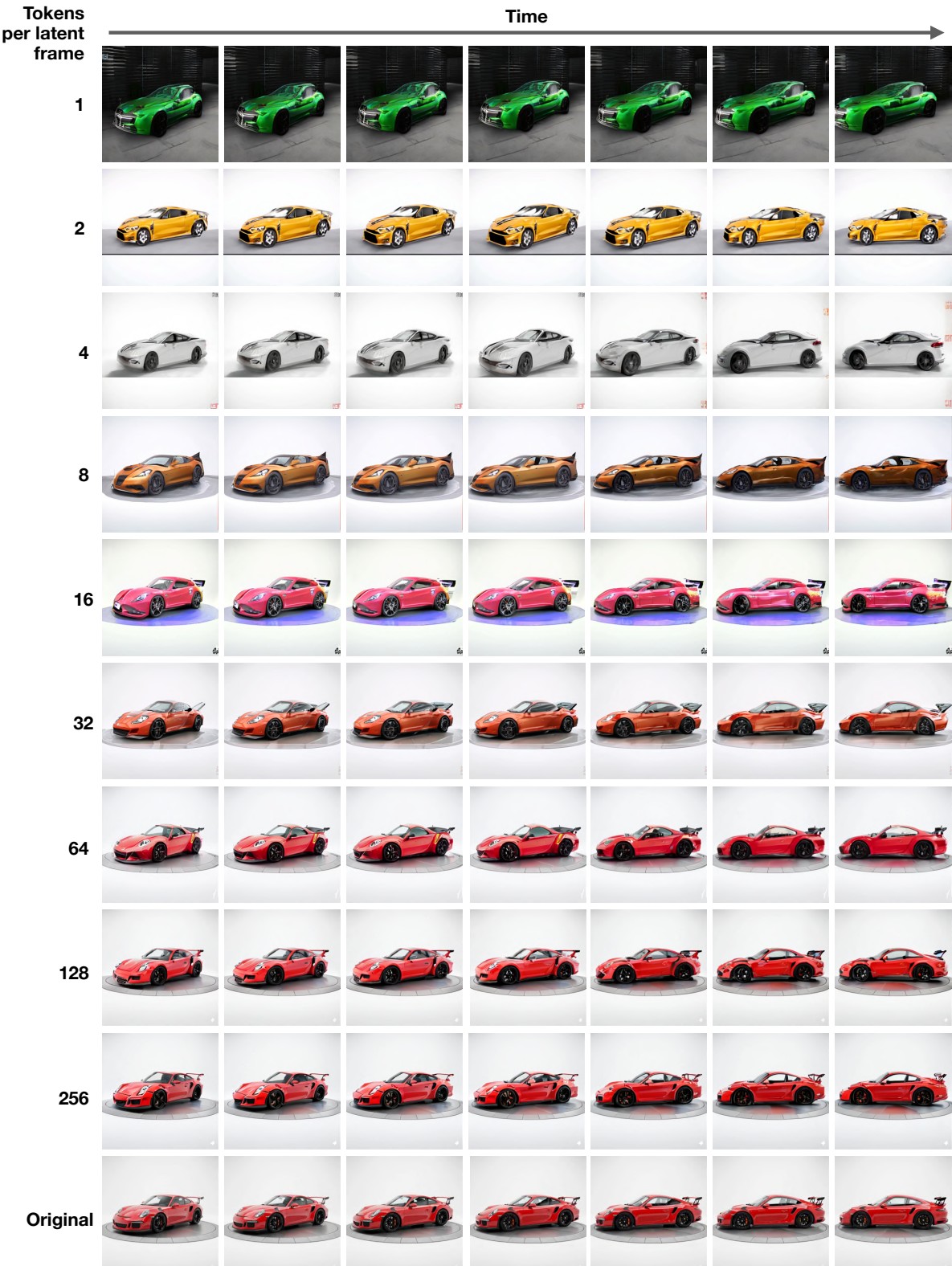

*Figure 17.* **VideoFlexTok reconstruction example**. From top to bottom each row corresponds to a video reconstructed using 1, 2, 4, ..., 256 tokens. The last row shows the original video.

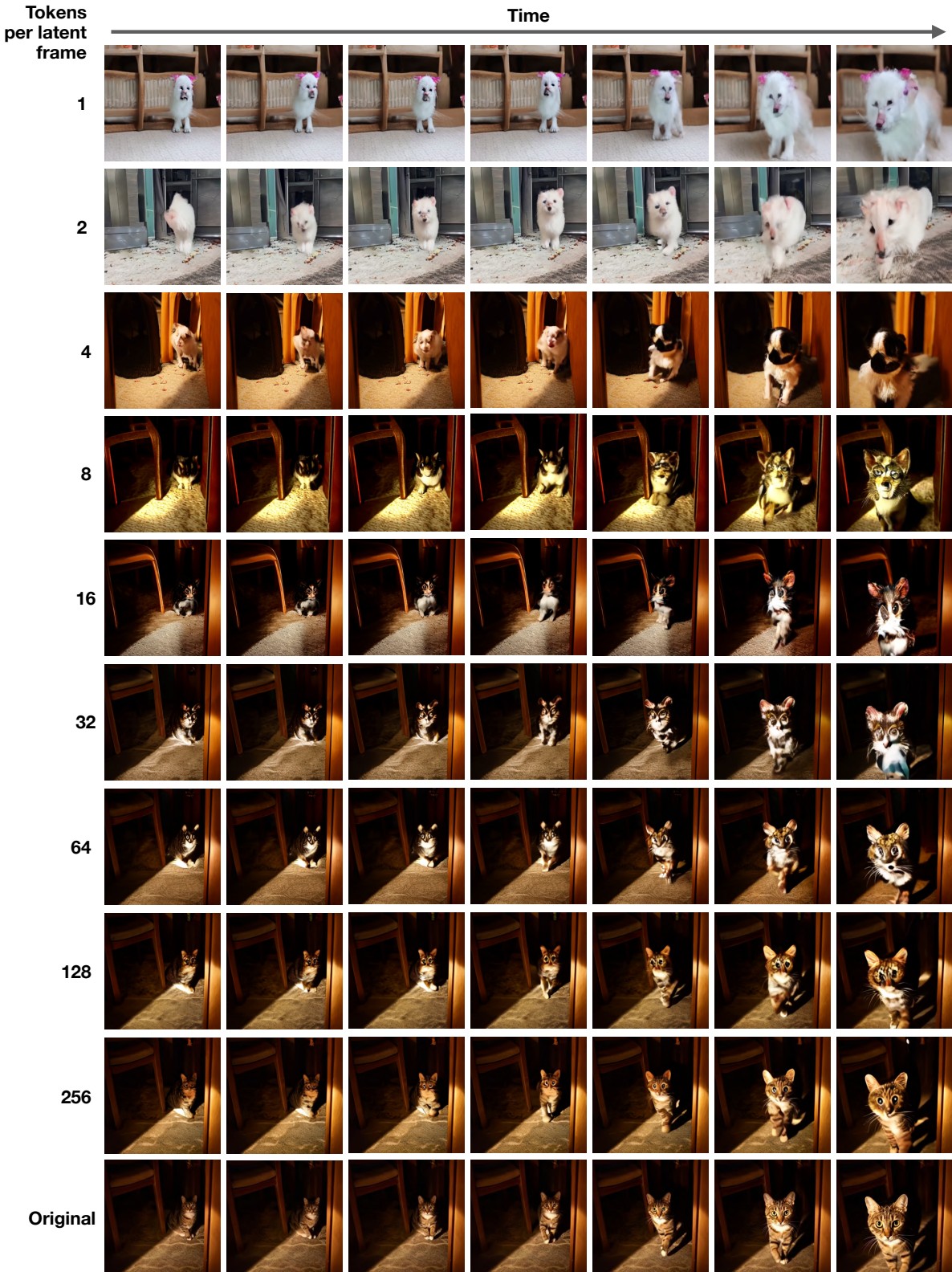

*Figure 18.* **VideoFlexTok reconstruction example**. From top to bottom each row corresponds to a video reconstructed using 1, 2, 4, . . . , 256 tokens. The last row shows the original video.

lowing changes. First, we use full attention in the decoder, which leads to more temporally-consistent reconstructions and better overall fidelity. Note that it is important to freeze the encoder during this stage, as using full decoder attention during encoder training leads to worse performance as we demonstrate in Section 4.5 and Table 3. Second, we introduce a frame-conditioning capability by randomly providing a clean VAE latent for the first latent frame, corresponding to the first frame due to the causal VAE, with probability $p = 0.5$. Similar to (Zheng et al., 2024), we find that even a short fine-tuning is enough to acquire this capability.

### F.2. Autoregressive model details

We provide a detailed overview of the architecture and training configuration for the autoregressive model in the class-to-video (Table 6) and text-to-video (Table 7) experiments. The AR models are causal decoder Transformers where the hidden size is tied to the depth via $w = 64d$, the number of attention heads equals the depth $d$, and the feed-forward layers apply an MLP ratio of $4$ relative to the attention dimension. We do not use $\mu$P (Yang et al., 2021) for the AR models, instead opting for scaling the learning rate inversely with the model width.

In the class conditioned setting, we mitigate overfitting to the relatively small Kinetics-600 dataset by applying dropout in the FFN, attention, and projection modules of the Transform with probability 0.1 and random cropping with resizing of the videos. All model sizes are trained for the same length 164B tokens.

In the text-to-video setting the training is not as data constrained, so we do not apply dropout in the Transformer. When scaling the AR model we follow compute-optimal training, training the different model sizes using the rule of thumb $D \approx 20N$ (Hoffmann et al., 2022). As we scale the number of training tokens we also scale the batchsize following a square-root relationship and rounding down to the nearest power of 2 (Zhang et al., 2025). We make the assumption for these compute-optimally trained models that the training of a decoder-only model on video data follows similar training token to parameter scaling as for a text-only model. To mitigate potential differences in training models on the two modalities we also train a model which is 4x over-trained relative to the Chinchilla compute optimal value.

### G. Reconstruction on MSR-VTT

Table 8 provides a comparison of VideoFlexTok for video reconstruction relative to common video tokenizers on the MSR-VTT dataset (Xu et al., 2016), which we use as an out-of-distribution dataset for our tokenizer trained on the large-scale Panda dataset. In these evaluations, we sample 17 frames at 4 FPS, 256 by 256 resolution from 5k samples from the MSR-VTT dataset. The VideoFlexTok d18-d28 version outperforms the baselines on reconstruction fidelity metrics such as rFID and rFVD with 1280 tokens per video clip. The VideoFlexTok is slightly worse on pixel-level reconstruction metrics such as MSE and MAE.

*Table 5.* `VideoFlexTok` **training settings.** Model and training configuration for different autoregressive Transformer sizes.

| VideoFlexTok configuration | d12-d12 (ablation setting) | d18-d18 | d18-d28 |
|---|---|---|---|
| Encoder depth $d_{enc}$ | 12 | 18 | 18 |
| Decoder depth $d_{dec}$ | 12 | 18 | 28 |
| Encoder dim. $w_{enc} = 64 d_{enc}$ | 768 | 1152 | 1152 |
| Decoder dim. $w_{dec} = 64 d_{dec}$ | 768 | 1152 | 1792 |
| Encoder Transformer parameters | 84.9M | 286.7M | 286.7M |
| Decoder Transformer parameters | 84.9M | 286.7M | 1.1B |
| Decoder adaLN (Peebles & Xie, 2023) parameters | 84.9M | 286.7M | 1.1B |
| Max. num. registers $K$ | $5 \times 256 = 1280$ | | |
| Encoder attention mask | Time-causal + causal over registers (ref. Section 3.1) | | |
| Decoder attention mask | Time-causal | | |
| Register nested dropout mode | Uniform from $\{1, 2, 4, \ldots, 256\}$ per latent frame (same for all) | | |
| FSQ (Mentzer et al., 2024) levels | $[8, 8, 8, 5, 5, 5]$ (64000 vocab. size) | | |
| VAE (Tang et al., 2024) | `vidtok_kl_causal_488_16chn` (Tang et al., 2024) | | |
| VAE channels | 16 | | |
| VAE downsampling factor (time×height×width) | $4 \times 8 \times 8$ | | |
| Patch size [time, height, width] | `[1, 1, 1]` | | `[1, 2, 2]` |
| REPA (Yu et al., 2025) layer | 1 | | |
| REPA (Yu et al., 2025) model | DINOv2-L (Oquab et al., 2024) | | |
| REPA (Yu et al., 2025) projection | Time-causal Transformer; depth 2; decoder width | | |
| REPA (Yu et al., 2025) loss weight | 1.0 | | |
| Training length ($n$ tokens) | 100B | 400B | 400B |
| Warmup length ($n$ tokens) | 4B | | |
| Warmup learning rate | `1e-6` | | |
| Learning rate schedule | Cosine decay | | |
| Optimizer | AdamW | | |
| Opt. momentum | $\beta_1, \beta_2 = 0.9, 0.99$ | | |
| Learning rate $\eta$ | `1.124e-3` | | |
| Batch size, samples | 512 | | 1280 |
| $\mu$P (Yang et al., 2021) base dim. | 256 | | |
| Gradient clipping norm | 1.0 | | |
| Decoder full-attention fine-tuning ($n$ tokens) | - | - | 400B |
| Dataset | Kinetics-600 | | Panda (30M subset) |
| Video resolution | `17x128x128` | | `17x256x256` |
| Data type | `bfloat16` (Burgess et al., 2019) | | |

*Table 6.* **Class-conditioned AR training settings.** Model and training configuration for different autoregressive Transformer sizes.

| Configuration | AR 49M | AR 85M | AR 201M | AR 393M | AR 679M | AR 1.33B | AR 2.29B |
|---|---|---|---|---|---|---|---|
| Num. non-embedding parameters | 49M | 85M | 201M | 393M | 679M | 1.33B | 2.29B |
| Decoder depth $d_{dec}$ | 10 | 12 | 16 | 20 | 24 | 30 | 36 |
| Decoder dim. $w_{dec}$ | 640 | 768 | 1024 | 1280 | 1536 | 1920 | 2304 |
| Cross-attention dim. | | | | n/a | | | |
| MLP ratio | | | | 4 | | | |
| Max. sequence length | | | | 1280 | | | |
| Attention mask | | | | Causal | | | |
| Vocab size | | | | 64,000 | | | |
| Feedforward activation | | | | SwiGLU (Shazeer, 2020) | | | |
| Positional encoding | | | | Learned embedding | | | |
| Conditioning dropout prob. | | | | 0.1 | | | |
| FFN, attn. and proj. dropout prob. | | | | 0.1 | | | |
| Training length ($n$ tokens) | | | | 164B | | | |
| Warmup length ($n$ tokens) | | | | $\approx$4.1B (2.5% of training tokens) | | | |
| Initial warmup learning rate | | | | $\eta \times 1e\text{-}3$ | | | |
| Learning-rate schedule | | | | Linear warmup + cosine decay | | | |
| Optimizer | | | | AdamW (Loshchilov & Hutter, 2017) | | | |
| Opt. momentum | | | | $\beta_1, \beta_2 = 0.9, 0.95$ | | | |
| Learning rate $\eta$ | 1.60e-3 | 1.33e-3 | 1.00e-3 | 8.00e-4 | 6.67e-4 | 5.33e-4 | 4.4e-4 |
| Final learning rate | | | | $\eta \times 1e\text{-}2$ | | | |
| Batch size | | | | 512 | | | |
| $\mu$P (Yang et al., 2021) base dim. | | | | n/a | | | |
| Weight decay | | | | 0.05 | | | |
| Weight-decay timescale $\tau_{iter}$ (Wang & Aitchison, 2025) | | | | n/a | | | |
| Gradient clipping norm | | | | 1.0 | | | |
| Dataset | | | | Kinetics-600 (Carreira et al., 2018) | | | |
| Video resolution | | | | `17x128x128` | | | |
| Augmentations | | | | `RandomResizedCrop` | | | |
| Data type | | | | `bfloat16` (Burgess et al., 2019) | | | |

*Table 7.* **Text-conditioned AR training settings.** Model and training configuration for different autoregressive Transformer sizes.

| Configuration | AR 400M | AR 1.1B | AR 2.0B | AR 5.2B | AR 5.2B (4×C) |
|---|---|---|---|---|---|
| Num. total parameters | 400M | 110M | 2.02B | 5.20B | 5.20B |
| Decoder depth $d_{dec}$ | 12 | 20 | 30 | 42 | 42 |
| Decoder dim. $w_{dec}$ | 768 | 1280 | 1920 | 2688 | 2688 |
| Cross-attention dim. | 12 | 20 | 30 | 42 | 42 |
| MLP ratio | | | 4 | | |
| Max. sequence length | | | 1280 | | |
| Attention mask | | | Causal | | |
| Vocab size | | | 64,000 | | |
| Feedforward activation | | | SwiGLU (Shazeer, 2020) | | |
| Positional encoding | | | Learned embedding | | |
| Training length ($n$ tokens) | 3.25B | 12B | 38B | 101B | 402B |
| Warmup length ($n$ tokens) | | | 2.5% of training tokens | | |
| Initial warmup learning rate | | | 1e-3 $\times \eta$ | | |
| Learning rate schedule | | | Cosine decay | | |
| Optimizer | | | AdamW (Loshchilov & Hutter, 2017) | | |
| Opt. momentum | | | $\beta_1, \beta_2 = 0.9, 0.95$ | | |
| Learning rate $\eta$ | 1.33e-3 | 8.00e-4 | 5.33e-4 | 3.79e-4 | 3.79e-4 |
| Final learning rate | | | $\eta \times$ 1e-2 | | |
| Batch size | 128 | 256 | 512 | 512 | 512 |
| $\mu$P (Yang et al., 2021) base dim. | | | n/a | | |
| Weight decay | | | 0.05 | | |
| Weight decay timescale $\tau_{iter}$ (Wang & Aitchison, 2025) | | | n/a | | |
| Gradient clipping norm | | | 1.0 | | |
| Dataset | | | Panda-70M (30M subset) | | |
| Video resolution | | | `17×256×256` | | |
| Augmentations | | | `RandomResizedCrop` | | |
| Data type | | | `bfloat16` (Burgess et al., 2019) | | |

*Table 8.* **Reconstruction metrics on MSR-VTT.** Evaluation is performed on 5k MSR-VTT videos (17 frames, 4 FPS, 256×256). All tokenizers use the same decoder architecture. [†] indicates `VideoFlexTok` results using 1280 tokens per video. [‡] marks results obtained from models evaluated at 128×128 input resolution.

| Tokenizer | # Tok. | MAE ($\downarrow$) | MSE ($\downarrow$) | PSNR ($\uparrow$) | LPIPS ($\downarrow$) | rFID ($\downarrow$) | rFVD ($\downarrow$) |
|---|---|---|---|---|---|---|---|
| VidTok FSQ (Tang et al., 2024) | 1280 | 0.0276 | 0.00270 | 25.64 | 0.0967 | 5.26 | 35.47 |
| LARP (Wang et al., 2025) | 1024 | 0.0476[‡] | 0.00760[‡] | 21.21[‡] | **0.1111**[‡] | 5.50[‡] | 28.33[‡] |
| Omnitokenizer (Junke et al., 2024) | 1280 | **0.0206** | **0.00100** | **30.01** | 0.1199 | 7.43 | 38.67 |
| Cosmos-DV (NVIDIA et al., 2025) | 1280 | 0.0287 | 0.00284 | 25.46 | 0.122 | 8.51 | 47.20 |
| `VideoFlexTok d18-d28` | 5–1280 | 0.0607[†] | 0.01079[†] | 19.67[†] | 0.18545[†] | **4.97**[†] | **20.53**[†] |

