# OpenReview forum: "VideoFlexTok: Flexible-Length Coarse-to-Fine Video Tokenization"
_ICML.cc/2026/Conference — ICML 2026 spotlight_

### Official Review · Reviewer_aXVV · 2026-03-07

**Soundness:** 3
**Presentation:** 3
**Significance:** 3
**Originality:** 2
**Overall Recommendation:** 4
**Confidence:** 3

**Summary:**

This paper proposes VideoFlexTok, a flexible video tokenizer that learns multiple token granularities to better support different video generation tasks. The method introduces adaptive token grouping and demonstrates improved performance and efficiency across several video generation benchmarks compared with existing tokenization approaches.

**Compliance With Llm Reviewing Policy:**

Affirmed.

**Final Justification:**

I agree with other reviewers' opinion that this paper can be viewed as a natural extension of FlexTok to the video domain, but provides some insights on meaningful adaptation. I will maintain my positive rating.

**Key Questions For Authors:**

In addition to the weaknesses part, I am also interested in the findings in Sec. 4.2, i.e., the first tokens tend to capture motion information. Could this observation be leveraged to design a temporally adaptive token allocation strategy, for example by preserving more tokens in early frames and fewer tokens in later frames to improve efficiency? It would be helpful if the authors could discuss whether such a design has been explored or provide preliminary analysis during rebuttal.

**Limitations:**

Yes

**Strengths And Weaknesses:**

# Strength
1. The paper is clearly written and well organized, with a clear motivation connecting tokenization flexibility to downstream video generation performance.
2. The experiments are fairly comprehensive and evaluate the approach on multiple benchmarks, providing useful empirical evidence for the proposed design.

# Weakness
1. While the idea of flexible token granularity is interesting, the conceptual novelty appears somewhat incremental since the method mainly extends existing tokenization frameworks with adaptive grouping rather than introducing a fundamentally new representation learning paradigm.
2. The experimental section could benefit from stronger ablations isolating the effects of codebook size/dim to better study the behavior of 1D flexible tokenization.
3. The work focuses on latent-space tokenization, but it is unclear how the proposed flexible tokenization strategy would behave in pixel-space modeling or whether similar benefits would hold when applied to pixel-based video generation models.

---

> ### Author Rebuttal · Authors · 2026-03-31
>
> We thank the reviewer for their time and thoughtful comments. We address the raised questions below.
>
> ---
> > conceptual novelty appears somewhat incremental
>
> While we inherit some ideas from prior work, our contribution is strong nonetheless. Importantly, **FlexTok does not tackle the challenge of aligning these ideas with the additional temporal structure of video data**. Our work provides novelty in the form of  1) "*well-executed design*" (peE5) of flexible representations for temporal video data that uniquely combines and adjusts established components, providing 2) "*empirical evidence for our design choices*" (aXVV) and "*extensive ablations*" (yBYg), and 3) "*rigorous empirical evaluation*" (yBYg) on downstream generative tasks.
> Specifically:
> 1. A naive extension of 1D tokenization (e.g., [7]) to video is suboptimal in downstream performance (Tab. 2) and does not allow streaming tokenization.
> 2. Using both time-causal encoder and decoder during training leads to better downstream generative performance vs. full attention (Tab. 3, Fig. 10).
> 3. We explored depth-first and time-first AR patterns over our temporal (time) coarse-to-fine (depth) 2D token structure (Tab. 4), with time-first enabling variable-length generation and demonstrating more efficient downstream modeling.
> 4. We provide rigorous evaluation at a large scale (using Panda-70M) across class-to-video and text-to-video downstream tasks, establishing clear scaling trends. We will also release our final models to the community.
>
> > it is unclear how the proposed flexible tokenization strategy would behave in pixel-space modeling
>
> This is a fair question, and you have highlighted a common limitation of video generative modeling which our work inherits. We primarily make this design choice of latent space modeling due to the difficulty of training the generative decoder part of the tokenizer directly in the pixel space. Our work does not attempt to address this limitation. Instead, we primarily focus on reorganizing the spatiotemporal video structure into a flexible-length coarse-to-fine tokens structure, which, in principle, should be applicable to the spatiotemporal pixel space. Future work could, for example, build upon the very recent work of [1], which develops an efficient image generative model acting directly in pixel space, to apply our proposed framework directly in pixel space.
>
> [1] Li, Tianhong, and Kaiming He. "Back to basics: Let denoising generative models denoise.”, arXiv’25
>
> > The experimental section could benefit from stronger ablations isolating the effects of codebook size/dim to better study the behavior of 1D flexible tokenization.
>
> Our work primarily focuses on exploring the tokenization structure aspect and developing flexible-length coarse-to-fine video representations. We include the ablations on the attention pattern (Tab. 3, Fig. 10), representation structure (Tab. 2), and different autoregressive patterns (Tab. 4). We, otherwise, default to reasonable choices following prior works, e.g., FlexTok and Cosmos tokenizers for the FSQ codebook dimensionality.
>
> > discussion on the temporally adaptive token allocation strategy
>
> This is a very interesting and natural idea reminiscent of classical video codecs! We, indeed, briefly explored this design during development. We share some of our early observations below.
>
> In this experiment, instead of sampling the same number of tokens for each latent frame in the nested dropout during training, we sample them independently for each latent frame. This allows allocating different token counts per frame. Specifically, for 5-latent frames videos (17 RGB frames), we explored patterns of the form Y+4X, independently varying the number of tokens for the first frame Y and subsequent frames X.
>
> We observe:
> - no improvement on average when applying the same token allocation structure to all samples compared to our standard uniform allocation.
> - improvements when choosing the allocation per-sample, but they are rather minor and apply to a minority of samples.
> - significant reconstruction quality drop in the low-token regime (e.g., 1-4 per frame) compared to our final design with uniform allocation.
>
> We suspect that the above limitations of this design might stem from a suboptimal sampling of the nested dropout masks during training, and exploring better sampling strategies in this setting can be an interesting future work. For example, in our final design, we sample uniformly from {1, 2, …, 128, 256} tokens during training instead of {1, 2, 3, …, 255, 256}, as the latter underrepresents the case of reconstructing from a single token (empirically), leading to poor reconstruction performance from a single token.

---

> > ### Author Rebuttal · Reviewer_aXVV · 2026-04-03
> >
> > I appreciate the authors' efforts during rebuttal, which has partially addressed my concerns. I will maintain my positive rating.
> >
> > Moreover, I think ablations on codebook size/dim, temporally adaptive token allocation strategy, will help to inspire the following works.

---

> > > ### Author Response · Authors · 2026-04-08
> > >
> > > We thank the reviewer for acknowledging our rebuttal.
> > >
> > > ---
> > >
> > > # General Response
> > >
> > > We thank the reviewers for their time and thoughtful comments. We appreciate that reviewers recognize the proposed “*thoughtful design choices*” (peE5) “*effectively validated by extensive ablations”* (yBYg),  “*rigorous*” (yBYg) and “*fairly comprehensive*” (aXVV) empirical evaluation, *“clear utility for scaling temporal context”* (yBYg), and “*convincing demonstration of the emergence of motion and semantics in the first tokens*” (peE5). Below we also provide a significance statement of potential broader impact.
> > >
> > > ## **Significance statement**
> > >
> > > We believe that the broader impact of our work goes beyond the extension of FlexTok to video data (still significant on its own). Importantly, **we propose a thoroughly evaluated design to handle an additional sequential structure in data, which is directly relevant to other domains** such as audio, time-series, or interleaved data. Prior flexible tokenization methods were mainly developed for static images and do not offer clear design guidance for these domains. VideoFlexTok addresses this gap. Specifically, we show that naively treating the sequential structure as just another axis (our non-causal 1D baseline) leads to subpar performance (Tab. 2 and 3, Fig. 10) and also rules out streaming applications.
> > >
> > > Furthermore, we see this work as **providing an important building block for more efficient world modeling and reasoning in latent spaces**, which require compact semantic representations of _sequential_ data. Specifically, we demonstrate downstream efficiency trends (Fig. 6) and the ability to model much longer sequences (empirically show ~5x in Sec. 4.4) at the same budget.

---

### Official Review · Reviewer_yBYg · 2026-03-12

**Soundness:** 3
**Presentation:** 3
**Significance:** 3
**Originality:** 3
**Overall Recommendation:** 4
**Confidence:** 3

**Summary:**

The authors present VideoFlexTok, a video tokenizer that replaces standard 3D spatiotemporal token grids with variable-length, coarse-to-fine token sequences to reduce the computational burden of video generation.
The proposed encoder uses nested dropout and a semantic bias loss to ensure the first few tokens capture high-level semantics and motion, while later tokens add fine-grained visual details.
A flow-based decoder is then used to reconstruct plausible videos from any given number of these tokens. By allowing downstream autoregressive models to use far fewer tokens, VideoFlexTok achieves comparable generation quality to fixed-grid baselines while using up to 10x less compute, effectively scaling to much longer video generation tasks.

**Compliance With Llm Reviewing Policy:**

Affirmed.

**Final Justification:**

The authors' rebuttal addresses my concerns. I will keep my score.

**Key Questions For Authors:**

- The paper notes that VideoFlexTok performs slightly worse on pixel-level metrics like MSE and MAE on the MSR-VTT dataset compared to the baselines. Given the importance of these metrics when evaluating latent reconstruction quality against standard baselines like MVAE or MMVAE, do you believe this is a fundamental trade-off of the semantic bias (REPA) or something that could be mitigated with a different quantization strategy?
- In Appendix C.2, the alignment score drops when generating longer sequences with the causal decoder, which is attributed to the limited capacity of the 200M AR model. Have you observed this instability vanishing completely with the 1.3B or larger models, or does the causal masking inherently make longer sequence prediction unstable?
- The inference cost analysis shows favorable GFLOPs scaling, but the method relies heavily on the flow decoder to fill in missing details when using shorter AR sequences. Could you provide a clearer breakdown of the wall-clock time trade-offs between AR generation steps and flow denoising steps in a standard deployment scenario?

**Limitations:**

No. The authors should consolidate the technical limitations, specifically the pixel-level reconstruction trade-offs and the reliance on flow decoder denoising steps.

**Strengths And Weaknesses:**

**Strengths**

- Shifting from rigid 3D spatiotemporal grids to a flexible, coarse-to-fine sequence is a highly practical and creative approach to video tokenization. The finding that the first few tokens naturally learn to capture scene geometry and motion without explicit label supervision is a compelling insight.
- The demonstrated computational efficiency gains address a major bottleneck, proving that comparable generation fidelity can be achieved using an autoregressive model that is ten times smaller. Showing that a 10-second video can be generated using eight times fewer tokens than standard baselines provides clear utility for scaling temporal context.
- The empirical evaluation is rigorous, utilizing a compute-aware scaling procedure to test the method across various model sizes and token budgets. Extensive ablations covering the REPA loss, attention mechanisms, and generation orders effectively validate the architectural claims.

**Weaknesses**

- The tokenizer trades off some pixel-perfect reconstruction capabilities for its semantic generation efficiency. In out-of-distribution evaluations, it performs slightly worse on low-level metrics like MSE and MAE compared to standard baselines.
- In the causal decoder ablation, alignment scores drop when generating longer sequences, which the authors attribute to the limited capacity of the 200M model. Providing evidence that this instability does not persist at larger scale configurations would strengthen the claims.
- Generating fewer tokens pushes more of the heavy lifting to the flow decoder to generate missing visual details. Discussing the trade-off in wall-clock inference time between autoregressive generation and flow denoising directly in the main text would clarify the overall efficiency gains.

---

> ### Author Rebuttal · Authors · 2026-03-31
>
> We thank the reviewer for their time and thoughtful comments. We address the raised questions below.
>
> > *Pixel-perfect reconstruction vs. generation performance trade-off.*
>
> This is an interesting questoin! We indeed observe a natural trade-off between compression and downstream generation efficiency. Importantly, **with a single VideoFlexTok model we can use more tokens for better reconstruction fidelity, or fewer tokens for more efficient downstream modeling**.
>
> Existing evidence and our experiments support this. [1] directly demonstrates this trade-off through scaling experiments, showing smaller generative models benefit from more compressed latents. [2, 3] demonstrate that modeling in semantic and, potentially, lossier spaces (e.g., SigLIP) is more efficient. Our Tab. 2 shows that fully-flat 1D video tokenization achieves better reconstruction performance but significantly lags behind the lossier but more structured 2D time-causal design in downstream generation.
>
> High reconstruction fidelity is, however, important for controllability. If the latent space is not expressive enough, a conditional generative model cannot express certain concepts. E.g., if tokens abstract away location information, the generative model will not be able to control for it. Different downstream applications (or even different prompts) will require different numbers of tokens to balance controllability and efficiency.
>
> The following table shows the rate-distortion for VideoFlexTok on Kinetics600. Using more tokens predictably improves reconstruction. We can vary the number of tokens with a single model at test time.
> |# Tok/Frame|MSE|LPIPS|rFVD|
> |---|---|---|---|
> |1|0.105|0.563|79.6|
> |4|0.075|0.465|56.2|
> |16|0.039|0.324|42.9|
> |64|0.020|0.213|28.8|
> |256|0.011|0.147|24.5|
>
> The table below shows downstream class-to-video performance across token budgets and AR model sizes. The small 0.3B AR model at 256 tokens produces low-fidelity videos (gFVD 177.9), but at 16 tokens achieves much better fidelity (gFVD 110.9) with a high classification score of 0.86, matching the larger 2.3B model at full budget. **The same VideoFlexTok model allows striking a better quality/controllability trade-off for different downstream compute budgets**.
>
> |AR size|Metric|1 tok|16 tok|256 tok|
> |---|---|---|---|---|
> |0.3B|gFVD|138.0|**110.9**|177.9|
> ||Cls. Score|0.69|**0.86**|0.82|
> |2.3B|gFVD|114.2|**86.5**|103.7|
> ||Cls. Score|0.60|0.82|**0.86**|
>
> [1] Ramanujan et al. "When worse is better: Navigating the compression-generation tradeoff in visual tokenization." NeurIPS'25
> [2] Zheng et al. "Diffusion transformers with representation autoencoders." ICLR'26
> [3] Tong et al. "Scaling Text-to-Image Diffusion Transformers with Representation Autoencoders." arXiv'26
>
> ---
>
> > *Alignment score instability with causal decoder for longer sequences (Appendix C.2). Does it vanish with 1.3B+ models?*
>
> Yes. As we show in Fig. 7 (bottom left), the alignment score of the 1.3B AR class-to-video model with our final time-causal VideoFlexTok increases and remains stable as we sample more tokens. We will fix the caption to specify the model size.
>
> ---
>
> > *AR vs. Flow inference cost*
>
> Fig. 12 compares total inference FLOPs across flow and AR configurations, showing that **generating ~64 AR tokens + 10-20 Flow steps yields the best performance (gFVD, ViCLIP) at any compute budget** and model size. We report FLOPs for a hardware-independent view.
>
> Below we provide inference time estimates across token budgets and Flow steps. We use 2B text-to-video AR model with KV-cache and d18-d28 VideoFlexTok on a single H100 (80GB). Similar to our inference FLOPs analysis, we find that, e.g., generating 64 AR tokens + 20 Flow steps is ~2.8x faster than genearating all 256 tokens while achieving better overall performance.
>
> |# Tokens|Metric|5 Flow|20 Flow|
> |---|---|---|---|
> |16|Time, s|4.79|10.4|
> ||gFVD|292.2|152.6|
> ||ViCLIP|0.1799|0.1859|
> |64|Time, s|13.54|**19.15**|
> ||gFVD|202.2|**158.4**|
> ||ViCLIP|0.1936|**0.1966**|
> |256|Time, s|48.55|54.16|
> ||gFVD|196.2|178.8|
> ||ViCLIP|0.1964|0.1987|
>
> Note that **wall-clock time depends heavily on the specific setting** (hardware, kernels, KV-cache management, batching). Both AR and Flow inference can be further optimized, e.g., by Flow distillation or MQA/MLA for the AR model. For example, as we increase the test batch size, the memory-bound AR model stays at ~37ms/token at BS=1-4, while the compute-bound Flow model cost grows linearly from 123ms to 374ms/step. We use the batch size of 4 in our other measurements above as it saturates VRAM, but using lower batch size would further skew the trend towards the Flow model.
>
> ---
>
> > The authors should consolidate the technical limitations, [...]
>
> We will extend our discussion of these points in accordance to our response above.

---

> > ### Author Rebuttal · Reviewer_yBYg · 2026-04-02
> >
> > This has resolved my concern. I will keep my score.

---

> > > ### Author Response · Authors · 2026-04-08
> > >
> > > We thank the reviewer for acknowledging our rebuttal.
> > >
> > > ---
> > >
> > > # General Response
> > >
> > > We thank the reviewers for their time and thoughtful comments. We appreciate that reviewers recognize the proposed “*thoughtful design choices*” (peE5) “*effectively validated by extensive ablations”* (yBYg),  “*rigorous*” (yBYg) and “*fairly comprehensive*” (aXVV) empirical evaluation, *“clear utility for scaling temporal context”* (yBYg), and “*convincing demonstration of the emergence of motion and semantics in the first tokens*” (peE5). Below we also provide a significance statement of potential broader impact.
> > >
> > > ## **Significance statement**
> > >
> > > We believe that the broader impact of our work goes beyond the extension of FlexTok to video data (still significant on its own). Importantly, **we propose a thoroughly evaluated design to handle an additional sequential structure in data, which is directly relevant to other domains** such as audio, time-series, or interleaved data. Prior flexible tokenization methods were mainly developed for static images and do not offer clear design guidance for these domains. VideoFlexTok addresses this gap. Specifically, we show that naively treating the sequential structure as just another axis (our non-causal 1D baseline) leads to subpar performance (Tab. 2 and 3, Fig. 10) and also rules out streaming applications.
> > >
> > > Furthermore, we see this work as **providing an important building block for more efficient world modeling and reasoning in latent spaces**, which require compact semantic representations of _sequential_ data. Specifically, we demonstrate downstream efficiency trends (Fig. 6) and the ability to model much longer sequences (empirically show ~5x in Sec. 4.4) at the same budget.

---

### Official Review · Reviewer_peE5 · 2026-03-12

**Soundness:** 3
**Presentation:** 3
**Significance:** 3
**Originality:** 2
**Overall Recommendation:** 5
**Confidence:** 4

**Summary:**

This paper proposes VideoFlexTok, a natural and solid extension of FlexTok for flexible-length video tokenization. It retains the core components of FlexTok (nested dropout, FSQ quantization, rectified flow decoder, REPA loss with DINOv2), while introducing video-specific modifications including time-causal attention in both encoder and decoder, a "time-first" token ordering strategy for autoregressive modeling, and a streaming decoding scheme with frame conditioning for arbitrary-length videos. Extensive experiments on class-to-video and text-to-video generation demonstrate that VideoFlexTok enables more efficient downstream generative modeling compared to standard 3D grid tokenization.

**Compliance With Llm Reviewing Policy:**

Affirmed.

**Final Justification:**

While the paper can be viewed as a natural extension of FlexTok to the video domain—somewhat limiting its novelty—it nonetheless represents a solid and meaningful adaptation. The work provides useful insights for the community and demonstrates clear value in extending the approach to video.

**Key Questions For Authors:**

Please refer to the weaknesses.

**Limitations:**

Please refer to the weaknesses.

**Strengths And Weaknesses:**

Strengths:
- The paper is well-motivated and clearly written. Rather than naively applying FlexTok to video, the authors make thoughtful video-specific design choices (time-causal attention, time-first ordering, streaming decoding) and provide clear justifications for each. The overall framework is coherent and well-engineered.

- The probing experiment that edits the first frame and conditions the decoder on only 1-2 tokens is well-designed. It provides an intuitive and convincing demonstration that the first tokens emergently capture motion and semantic information while abstracting away low-level details.

- Once well trained and converged, VideoFlexTok provides an efficient and highly compressed representation for videos. These representations can be effectively utilized to reduce the training cost of downstream tasks.

Weaknesses:
- The technical novelty beyond FlexTok is limited. The core methodology is inherited entirely from FlexTok, and the video-specific modifications, while well-executed, are individually well-established.

- There is an inconsistency regarding the decoder attention mechanism. The main paper and Table 3 both claim that time-causal self-attention in the decoder is optimal. However, the appendix describes an additional decoder fine-tuning stage where full attention replaces time-causal attention to improve reconstruction quality and temporal consistency. Could the authors clarify on this?

- The paper primarily compares against 3D grid tokenizers, but comparisons with recent continuous video tokenizers that also achieve high compression rates (e.g., StepVideo VAE, LTX Video VAE, DC-AE-V) would better contextualize VideoFlexTok's compression-quality tradeoff. Besides, the training cost of the tokenizer itself should be revealed more.

---

> ### Author Rebuttal · Authors · 2026-03-31
>
> We thank the reviewer for their time and thoughtful comments. We address the raised questions below.
> > The technical novelty beyond FlexTok is limited. [...]
>
> While we inherit some ideas from prior work, our contribution is strong nonetheless. Importantly, **FlexTok does not tackle the challenge of aligning these ideas with the additional temporal structure of video data**. Our work provides novelty in the form of  1) "*well-executed design*" (peE5) of flexible representations for temporal video data that uniquely combines and adjusts established components, providing 2) "*empirical evidence for our design choices*" (aXVV) and "*extensive ablations*" (yBYg), and 3) "*rigorous empirical evaluation*" (yBYg) on downstream generative tasks.
> Specifically:
> 1. A naive extension of 1D tokenization (e.g., [7]) to video is suboptimal in downstream performance (Tab. 2) and does not allow streaming tokenization.
> 2. Using both time-causal encoder and decoder during training leads to better downstream generative performance vs. full attention (Tab. 3, Fig. 10).
> 3. We explored depth-first and time-first AR patterns over our temporal (time) coarse-to-fine (depth) 2D token structure (Tab. 4), with time-first enabling variable-length generation and demonstrating more efficient downstream modeling.
> 4. We provide rigorous evaluation at large scale (using Panda-70M) across class-to-video and text-to-video downstream tasks establishing clear scaling trends. We will also release our final models to the community.
>
> ---
> > There is an inconsistency regarding the decoder attention mechanism. [...]
>
> When comparing full and time-causal attention in the decoder, we proceed in the two-stage manner: 1) train the full model with full/time-causal attention and 2) fine-tune *only the decoder* with full attention. Note that the encoder, and, hence, learned representations stay fixed during stage 2) and only the decoder is fine-tuned as we find it to result in a better readout performance (lower r/gFVD compared to no stage 2)). We perform the stage 2) for the full attention to make the two runs comparable in terms of the total number of optimization steps and tokens seen. Therefore, the time-causal decoder is important during the encoder training as a representation learning objective as it leads to better downstream performance (lower gFVD, higher alignment score). We will clarify this in the final draft.
>
> ---
> > [...] comparisons with recent continuous video tokenizers that also achieve high compression rates [...]
>
> Our work specifically explores flexible tokenization within discrete tokenization for video generation. Discrete tokenizers directly benefit from the simplicity and scalability of GPT-like autoregressive modeling [2], optimized LLM serving infrastructure [5], and multi-modal modeling [3, 4]. That said, VideoFlexTok's ideas can be extended to continuous tokens, e.g., similar to [6].
>
> Your comments on compression rate are worth addressing. While continuous VAE latents often achieve higher compression at the same or lower *number of tokens*, these tokens use more bytes per token due to being *8/16-dim float vectors*, which accounts for their superior reconstruction.
>
> For example:
> - StepVideo VAE (8x16x16 compression, 16-dim latent) compresses 17x256x256 RGB input into 3x16x16=768 16-dim FP32 tokens: **49152 bytes.**
> - VideoFlexTok compresses the same video into 5x1 to 5x256 discrete tokens with 64K vocabulary size: **20-2560 bytes, giving 19-2457x better compression rate.**
>
> Importantly, unlike VAE tokenizers relying on fixed 3D spatiotemporal grids, **VideoFlexTok enables flexibility over the compression rate** while prioritizing semantics and motion in early tokens. Please also refer to our response to yBYg, where we show that more tokens predictably improve reconstruction and this flexibility enables more efficient downstream modeling.
>
> ---
> > Besides, the training cost of the tokenizer itself should be revealed more.
>
> We use the following main model in the paper:
> - K600 VideoFlexTok d18-d18: 169 hours on 64 H100s
> - Panda VideoFlexTok d18-d28: 162 hours on 256 H100s
>
> We will add these to Tab. 5 alongside training settings. We also note that we will release these models.
>
> *References:*
>
> [1] Yu, Lijun, et al. "Language Model Beats Diffusion--Tokenizer is Key to Visual Generation.", ICLR'24
>
> [2] Sun, Pieze, et al. "Autoregressive Model Beats Diffusion: Llama for Scalable Image Generation."
>
> [3] Yang, Ling, et al. "Mmada: Multimodal large diffusion language models.", NeurIPS'25
>
> [4] Bachmann, Roman, et al. "4m-21: An any-to-any vision model for tens of tasks and modalities.", NeurIPS'24
>
> [5] Kwon, Woosuk, et al. "Efficient memory management for large language model serving with paged attention." SOSP'23 (https://github.com/vllm-project/vllm)
>
> [6] Wen, Xin, et al. "" Principal Components" Enable A New Language of Images.", ICCV'25
>
> [7] Wang, Hanyu, et al. "Larp: Tokenizing videos with a learned autoregressive generative prior.", ICLR'25

---

> > ### Author Rebuttal · Reviewer_peE5 · 2026-04-03
> >
> > I thank the authors for their thorough rebuttal, which has addressed my previous concerns. While the paper can be viewed as a natural extension of FlexTok to the video domain—somewhat limiting its novelty—it nonetheless represents a solid and meaningful adaptation. The work provides useful insights for the community and demonstrates clear value in extending the approach to video.
> >
> > Based on the clarifications and additional details provided, I am inclined to raise my score to 5.
> >
> > For further strengthening the impact of the work, it would be highly beneficial if the authors could include more detailed implementation specifics and consider open-sourcing the code and checkpoints, similar to FlexTok.

---

> > > ### Author Response · Authors · 2026-04-08
> > >
> > > We thank the reviewer for acknowledging our rebuttal and increasing the score.
> > >
> > > > For further strengthening the impact of the work, it would be highly beneficial if the authors could include more detailed implementation specifics and consider open-sourcing the code and checkpoints, similar to FlexTok.
> > >
> > > We will provide more details and open-source both.
> > >
> > > ---
> > >
> > > # General Response
> > >
> > > We thank the reviewers for their time and thoughtful comments. We appreciate that reviewers recognize the proposed “*thoughtful design choices*” (peE5) “*effectively validated by extensive ablations”* (yBYg),  “*rigorous*” (yBYg) and “*fairly comprehensive*” (aXVV) empirical evaluation, *“clear utility for scaling temporal context”* (yBYg), and “*convincing demonstration of the emergence of motion and semantics in the first tokens*” (peE5). Below we also provide a significance statement of potential broader impact.
> > >
> > > ## **Significance statement**
> > >
> > > We believe that the broader impact of our work goes beyond the extension of FlexTok to video data (still significant on its own). Importantly, **we propose a thoroughly evaluated design to handle an additional sequential structure in data, which is directly relevant to other domains** such as audio, time-series, or interleaved data. Prior flexible tokenization methods were mainly developed for static images and do not offer clear design guidance for these domains. VideoFlexTok addresses this gap. Specifically, we show that naively treating the sequential structure as just another axis (our non-causal 1D baseline) leads to subpar performance (Tab. 2 and 3, Fig. 10) and also rules out streaming applications.
> > >
> > > Furthermore, we see this work as **providing an important building block for more efficient world modeling and reasoning in latent spaces**, which require compact semantic representations of _sequential_ data. Specifically, we demonstrate downstream efficiency trends (Fig. 6) and the ability to model much longer sequences (empirically show ~5x in Sec. 4.4) at the same budget.

---

### Decision · Program_Chairs · 2026-04-30

**Decision:**

Accept (spotlight)

**Comment:**

3D spatiotemporal grids with flexible, coarse-to-fine token sequences for efficient video generation. By leveraging time-causal attention and a flow-based decoder, the framework enables downstream autoregressive models to achieve up to a 10x compute reduction while maintaining high generation quality. Although reviewers noted that the conceptual novelty is a natural progression from its image-based predecessor, they praised the thoughtful video-specific adaptations. The rebuttal successfully clarified technical details regarding decoder attention and inference trade-offs, leading to a consensus that the work provides a solid, practical solution to the computational bottlenecks of video tokenization. All the reviewers gave positive ratings after rebuttal. Please considering include the rebuttal experiments in camera-ready.